DOI: 10.1038/s41467-017-02170-3　　**OPEN**

# Auditory closed-loop stimulation of EEG slow oscillations strengthens sleep and signs of its immune-supportive function

Luciana Besedovsky[1], Hong-Viet V. Ngo[1,2], Stoyan Dimitrov[1,3,4], Christoph Gassenmaier[5], Rainer Lehmann[3,4,6] & Jan Born[1,3,4,7]

Sleep is essential for health. Slow wave sleep (SWS), the deepest sleep stage hallmarked by electroencephalographic slow oscillations (SOs), appears of particular relevance here. SWS is associated with a unique endocrine milieu comprising minimum cortisol and high aldosterone, growth hormone (GH), and prolactin levels, thereby presumably fostering efficient adaptive immune responses. Yet, whether SWS causes these changes is unclear. Here we enhance SOs in men by auditory closed-loop stimulation, i.e., by delivering tones in synchrony with endogenous SOs. Stimulation intensifies the hormonal milieu characterizing SWS (mainly by further reducing cortisol and increasing aldosterone levels) and reduces T and B cell counts, likely reflecting a redistribution of these cells to lymphoid tissues. GH remains unchanged. In conclusion, closed-loop stimulation of SOs is an easy-to-use tool for probing SWS functions, and might also bear the potential to ameliorate conditions like depression and aging, where disturbed sleep coalesces with specific hormonal and immunological dysregulations.

[1] Institute of Medical Psychology and Behavioral Neurobiology, University of Tübingen, Otfried-Müller-Straße 25, 72076 Tübingen, Germany. [2] School of Psychology, University of Birmingham, Edgbaston, Birmingham B15 2TT, UK. [3] German Center for Diabetes Research, Otfried-Müller-Straße 10, 72076 Tübingen, Germany. [4] Institute for Diabetes Research and Metabolic Diseases of the Helmholtz Center Munich at the University of Tübingen, Otfried-Müller-Straße 10, 72076 Tübingen, Germany. [5] Department of Internal Medicine IV, University of Tübingen, Otfried-Müller-Straße 10, 72076 Tübingen, Germany. [6] Central Laboratory, Department for Diagnostic Laboratory Medicine, University of Tübingen, Hoppe-Seyler-Straße 3, 72076 Tübingen, Germany. [7] Centre for Integrative Neuroscience, University of Tübingen, Otfried-Müller-Straße 25, 72076 Tübingen, Germany. Correspondence and requests for materials should be addressed to J.B. (email: jan.born@uni-tuebingen.de)

Sleep is crucial for general health, as demonstrated by epidemiological and experimental studies[1,2]. Sleep is a unique behavioral state that affects most, if not all, body functions, including the endocrine and immune systems[3,4]. The immune-supportive function of sleep is thought to be primarily conveyed by slow wave sleep (SWS)[3], which is the deepest stage of non-rapid eye movement (NonREM) sleep. SWS is hallmarked by slow waves in the electroencephalogram (EEG), which have a frequency of 0.5–4 Hz and include the slow oscillation (SO) frequencies ≤1 Hz. Slow wave activity (i.e., the spectral power in the frequency range of 0.5–4 Hz) is associated with the coordinate release of immune-active hormones, specifically with a suppression of cortisol and an increase in prolactin, growth hormone (GH), and aldosterone levels, which provides an optimal endocrine milieu for supporting adaptive immune functions[3,5–7]. Supporting this view, SO activity as well as the accompanying hormonal changes in prolactin, GH, and cortisol levels on the night after a vaccination against hepatitis A virus were highly predictive ($r \geq 0.71$) of the antigen-specific immune response to the vaccine, measured up to 1 year later[8]. However, so far, evidence for the relationship between slow wave or SO activity and the endocrine and immune functions of interest is solely based on correlational findings and, to the best of our knowledge, there is no direct proof of a causal role of SOs and SWS in this relationship. There are a few studies showing that selective suppression of SWS can affect peripheral functions, like glucose tolerance and blood pressure[9–11]. However, there have been no attempts to specifically strengthen SWS or SOs to boost its accompanying endocrine and immunological changes.

We have previously shown in healthy humans that closed-loop stimulation of SOs, by delivering tones in synchrony with the upstate of online detected SOs, can induce trains of high-amplitude SOs, thereby deepening NonREM sleep and improving its memory-forming function[12,13]. Here we asked whether closed-loop auditory stimulation of SOs can also enhance the endocrine characteristics of SWS that are considered to mediate the immune-supportive effects of sleep. As a read-out of immune effects, we assessed T and B lymphocyte numbers in blood because the redistribution of these cells from the circulation to lymphoid tissues is thought to be a central mechanism underlying the immune-enhancing effect of sleep[3,14]. We show that selectively enhancing SOs through auditory stimulation intensifies the immune-supportive hormonal milieu present during SWS (mainly by further reducing cortisol and increasing aldosterone levels) and supports the extravasation of T and B lymphocytes. The findings suggest a causal role for SOs in the regulation of endocrine activity and adaptive immune functions.

## Results

**Effects of auditory stimulation on EEG characteristics.** Fourteen healthy men spent two experimental nights (Stimulation vs. Sham stimulation) in the sleep laboratory, where they were allowed to sleep from 2300 to 0700 hours. In the Stimulation condition, closed-loop auditory stimulation of SOs was applied for 120 min, starting with the beginning of consolidated NonREM sleep, whereas in the Sham condition, time points of potential stimulation were marked without applying the stimulation (Fig. 1a, see "Methods" section for details). Blood for assessment of hormone concentration and T and B lymphocyte counts was collected before the start of stimulation and repeatedly across the night.

On average, the stimulation started 49 min ($\pm$6.86 s.e.m.) after lights were turned off and on average 172 ($\pm$24.3 s.e.m.) acoustic stimulations (each including two tones) were delivered in the

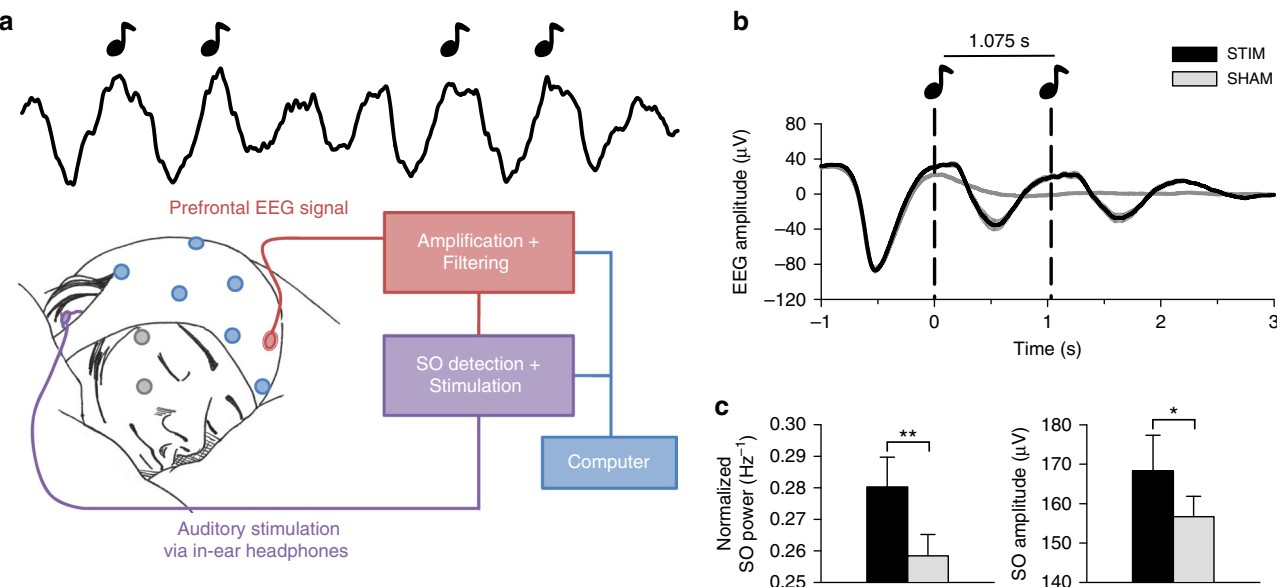

**Fig. 1** Auditory stimulation phase locked to endogenous SO peaks boosts SO activity. **a** Setup: Upon online detection of an endogenous SO in the frontal EEG signal during non-rapid eye movement (NonREM) sleep, two tones (50 ms, pink noise, 50 dB SPL) were delivered via in-ear headphones with an inter-stimulus interval of 1.075 s to coincide with two consecutive SO peaks. In the Sham condition, time points of stimulation were marked, but no stimuli were presented. See "Methods" section for further details. Artwork by H.-V.V.N. **b** Mean ($\pm$s.e.m.) EEG signal recorded from a frontal (Fz) electrode position during NonREM sleep (S2, S3, and S4) in the 120-min stimulation period, time-locked to the first of the two tones ($t = 0$) for the Stimulation (STIM, black) and Sham condition (SHAM, gray). **c** Mean ($\pm$s.e.m.) normalized spectral power in the SO peak frequency band (0.8–1.1 Hz) and SO amplitude recorded from electrode position Fz and determined for NonREM sleep epochs of the 120-min stimulation period. The average number of NonREM sleep epochs used for this calculation was 158 and 169, respectively, for the Stimulation and the Sham condition. (There was no significant difference in the number of epochs between conditions, $p = 0.123$). For normalization, individual spectra were divided by the cumulative power (up to 30 Hz). $**p < 0.01$, $*p < 0.05$, for pairwise comparisons between the Stimulation condition (STIM, black) and the Sham condition (SHAM, gray) with paired $t$ tests, two-sided. $n = 14$

**Table 1 Sleep architecture for entire night**

|  | STIM | | SHAM | | p values |
|---|---|---|---|---|---|
|  | **Means** | **s.e.m.** | **Means** | **s.e.m.** |  |
| In min |  |  |  |  |  |
| WASO | 14.39 | 3.72 | 13.54 | 3.09 | 0.75 |
| S1 | 49.86 | 5.34 | 56.25 | 10.10 | 0.44 |
| S2 | 235.96 | 8.20 | 231.61 | 7.44 | 0.63 |
| S3 | 51.86 | 5.76 | 52.18 | 4.23 | 0.94 |
| S4 | 25.29 | 4.45 | 24.43 | 3.64 | 0.80 |
| SWS | 77.14 | 7.40 | 76.61 | 5.10 | 0.90 |
| REM | 94.36 | 7.86 | 89.82 | 7.30 | 0.21 |
| In % |  |  |  |  |  |
| WASO | 3.08 | 0.80 | 2.90 | 0.67 | 0.76 |
| S1 | 10.56 | 1.13 | 12.02 | 2.17 | 0.41 |
| S2 | 50.04 | 1.76 | 49.51 | 1.59 | 0.78 |
| S3 | 10.96 | 1.20 | 11.15 | 0.90 | 0.83 |
| S4 | 5.37 | 0.95 | 5.22 | 0.78 | 0.85 |
| SWS | 16.33 | 1.55 | 16.38 | 1.09 | 0.96 |
| REM | 19.99 | 1.65 | 19.20 | 1.56 | 0.27 |

Mean (±s.e.m.) of absolute and percent of time spent in the different sleep stages
WASO, wake after sleep onset; S1, S2, S3, and S4, NonREM sleep stages 1–4; SWS, slow wave sleep (i.e., the sum of S3 and S4); REM, rapid eye movement sleep
p values refer to two-sided pairwise comparisons between the Stimulation (STIM) and the Sham condition with paired t tests. n = 14

120-min interval. Confirming our previous studies[12], closed-loop auditory stimulation distinctly increased SO activity during NonREM sleep in the stimulation interval: compared with the Sham condition, stimulation significantly increased EEG power density in the SO peak frequency (~0.9 Hz, corresponding to the 1.075 s inter-stimulus interval of the applied two tones per stimulation) as well as the amplitude of SOs ($p = 0.006$ and $p = 0.047$, respectively, for electrode position Fz; Fig. 1b, c; Cohen's $d = 0.71$ and 0.42, respectively). Other frequencies remained unchanged and SO activity was only affected during the stimulation interval but not afterwards (Supplementary Figs. 1 and 2). The absolute and percent time spent in the different sleep stages remained unaffected by the stimulation (Table 1), with this pattern overall confirming the specificity of this method selectively enhancing SOs. SWS duration in minutes or as a percent of total sleep time was also not affected in separate analyses of the stimulation interval and the post-stimulation interval (27% ±3.3 s.e.m. and 10% ±1.7 s.e.m., respectively, for the Stimulation condition and 30% ±4.2 s.e.m. and 9% ±1.4 s.e.m., respectively, for the Sham condition, $p > 0.351$).

**Effects of stimulation on endocrine and immune parameters.** Stimulation distinctly reduced blood cortisol concentrations during the first hour after stimulation onset ($p = 0.048$, Fig. 2a, b and Supplementary Fig. 3a), although levels were already rather low during this early night interval. The reduction was visible already 5 min post-stimulation onset and averaged 15% in the first hour of stimulation (Fig. 2a). Aldosterone levels were significantly increased in the Stimulation compared to the Sham condition during the fourth hour after stimulation onset ($p = 0.028$; Fig. 2c and Supplementary Fig. 3b). An increase in aldosterone levels during the eighth hour and in prolactin levels during the third hour after stimulation onset approached significance ($p ≤ 0.075$; Fig. 2c, d and Supplementary Fig. 3b, c), with exploratory analyses revealing enhanced prolactin levels after stimulation at two of the 15-min samplings (150 min and 165 min post-stimulation onset, respectively, $p ≤ 0.050$). GH remained unchanged (Supplementary Fig. 3d). Effect sizes ($r$ for nonparametric tests) were −0.37 for cortisol, 0.49 and 0.40,

respectively, for aldosterone, and 0.38 for prolactin, which reflect medium-sized effects. Blood T and B lymphocyte counts were acutely reduced 3 h post-stimulation onset ($p = 0.011$ and $p = 0.021$, respectively; Fig. 3). Respective effect sizes were $r = −0.60$ for T cells and $r = −0.54$ for B cells, which are considered large effect sizes.

**Regression analyses.** We performed explorative regression analyses to further examine associations between SO activity and concentrations of the endocrine and immune parameters of interest, and also to assess possible confounding effects of the auditory stimuli per se (see "Methods" section for details). SO activity was a significant predictor of cortisol levels ($β = −0.397$, $p = 0.019$) and, with a time lag of 2 h, of aldosterone levels, with the time lag reflecting the delayed impact of stimulation on this parameter ($β = 0.577$, $p = 0.045$). We did not find correlations between SO activity and lymphocytes, possibly because effects on these parameters were less direct. The number of applied auditory stimuli significantly predicted SO activity ($β = 0.420$, $p = 0.004$), but was not associated with endocrine/immune parameters, which rules out that the auditory stimuli per se substantially contributed to the hormonal and immunological effects of the SO stimulation.

**Discussion**

We show here that deepening sleep by EEG closed-loop auditory stimulation of SOs robustly decreases levels of the antiinflammatory hormone cortisol, increases levels of aldosterone and tends to increase also prolactin levels, thus enforcing the immune-supportive hormonal milieu that is unique to SWS[3]. In addition, stimulation markedly reduced T and B lymphocytes numbers in blood. Previous studies in humans have demonstrated that NonREM sleep and particularly SWS is associated with a unique pattern of endocrine activity comprising reduced cortisol concentrations, as well as increased aldosterone, prolactin, and GH levels[5,15–17]. The secretion of these hormones is temporally associated with slow wave activity[5–7,18]. Considering that our method is highly specific in selectively enhancing SOs without affecting other sleep parameters, our findings suggest a causal role of SOs in supporting the overall pro-inflammatory hormonal milieu characterizing SWS.

The lack of effect of the SO stimulation on GH was surprising, given that its release shows a clear association with EEG slow waves and is regulated most strongly by sleep[6,19,20]. However, the increase in SO activity by the stimulation was only moderate in size (probably due to our subjects being young healthy men with already deep sleep) and, thus, this increase might not have been large enough to affect GH. Nevertheless, the increase in SO activity was comparable in size to that of our previous study[12] and strong enough to affect other parameters. The lack of effect on GH might therefore rather reflect a ceiling effect for GH as the release of this peptide hormone is already at a diurnal maximum during early nocturnal SWS in healthy humans. Thus, an enforcing effect of auditory stimulation of SOs may become apparent only in conditions of shallowed sleep and accompanying reductions in GH levels, such as during aging[21,22]. Another explanation could be that the release of GH is not controlled specifically by slow waves in the SO frequency range (i.e., ≤1 Hz), but by higher frequencies in the delta range (1–4 Hz), which were not altered by our approach. On the other hand, there is also evidence that GH secretion is not directly depending on SWS but rather on sleep onset because selective SWS deprivation did not affect GH levels and also dissociations between the occurrence of SWS and GH secretion have been found[23–25]. Based on these findings, it has been suggested that there is a neuronal mechanism

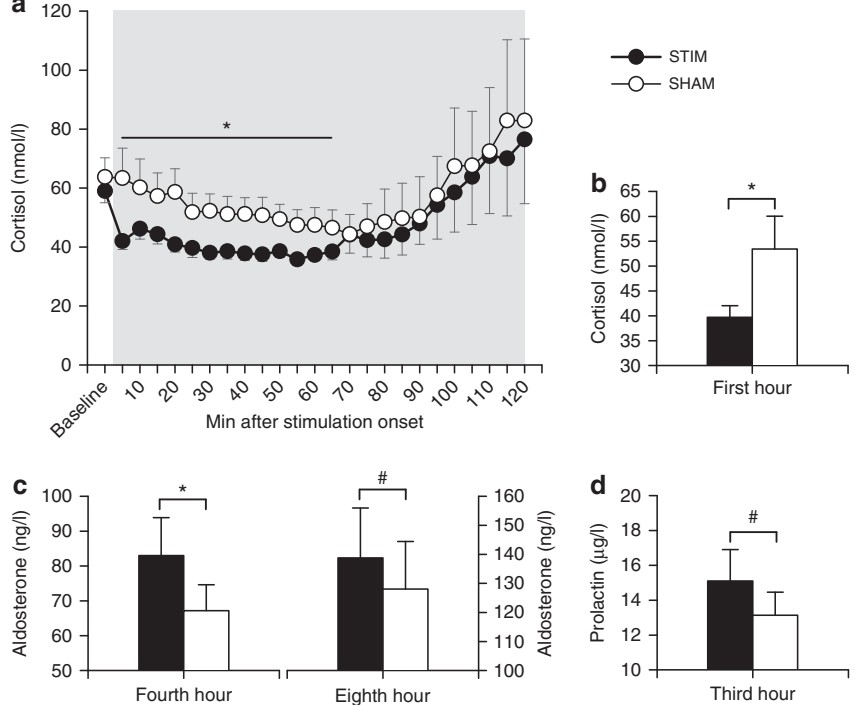

**Fig. 2** Impact of auditory SO stimulation on cortisol, aldosterone, and prolactin levels. Means (±s.e.m.) of cortisol (**a**, **b**), aldosterone (**c**), and prolactin (**d**) levels calculated for 1-h bins. Gray area represents the 120-min stimulation period. *$p < 0.05$, #$p < 0.1$ for pairwise comparisons between the Stimulation condition (STIM, black) and the Sham condition (SHAM, white) with Wilcoxon tests, two-sided. $n = 10$–14 (see "Methods" section for exact numbers)

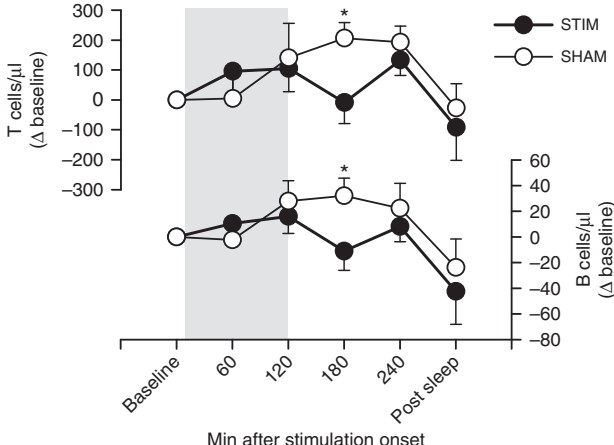

**Fig. 3** Impact of auditory SO stimulation on lymphocyte counts. Means (±s.e.m.) of circulating T and B cell numbers (shown as difference from baseline). Gray area represents the 120-min stimulation period. *$p < 0.05$ for pairwise comparisons between the Stimulation condition (STIM, black) and the Sham condition (SHAM, white) with Wilcoxon tests, two-sided. $n = 9$

coupling the onset of sleep, SWS and GH secretion, leading to the typically observed association between SWS and GH levels, in the absence of a direct causal effect of this sleep stage on GH secretion[23,25]. This concept would conclusively explain why we did not observe changes in GH levels following selective enhancement of SOs.

The reduction in cortisol levels was visible already 5 min following stimulation onset. This fast effect points to a regulation by changes in autonomic nervous system activity[26], as SOs are known to also affect activity of brainstem nuclei, like the locus coeruleus, that are centrally involved in autonomic nervous

system regulation[27,28]. In contrast to cortisol, aldosterone was affected with a delay of 3–4 h (after stimulation onset). Although the release of both corticosteroids aldosterone and cortisol from the adrenal cortex can be stimulated by adrenocorticotropic hormone (ACTH), aldosterone secretion during SWS is mainly regulated by renin[7], leading to a unique uncoupling of cortisol and aldosterone secretion during this sleep stage. The effect of SO stimulation on prolactin levels developed gradually (Supplementary Fig. 3c) and failed to reach significance after clustering data of the frequent blood samplings into 1-h bins. The pituitary release of prolactin is mainly regulated in a negative, i.e., suppressive manner by dopaminergic activity of the arcuate nucleus of the hypothalamus. Interestingly, this activity exhibits an endogenous rhythm of 0.05–4 Hz[29] and, thus, might be sensitive to exogenous stimulation of oscillations in the same frequency range. This idea is supported by our findings. However, the effect of the stimulation on prolactin was not robust, and also in light of conflicting findings as to an association of prolactin release with specific sleep stages[20], the role of SOs in the regulation of this hormone needs to be further scrutinized. Overall, the exact mechanisms of the stimulation-induced changes in endocrine activity are clearly in need of further investigation.

SO stimulation, with a delay of about 3 h, decreased numbers of T and B lymphocytes in blood. The decrease is in line with findings of a general acute reduction in lymphocyte numbers during sleep when compared to nocturnal wakefulness[30,31]. The effect of SO stimulation emerging with some delay, indeed, mimics the rather slow temporal dynamics of the influences of sleep on numbers of circulating lymphocytes. These effects are most likely originating from hormonal changes, rather than reflecting peripheral nervous system regulation, which is much faster acting (in the minute range). In fact, stimulation-induced increases in aldosterone and prolactin levels could have contributed to the delayed changes in lymphocyte counts, as both hormones exert specific influences on lymphocyte migration[32–36],

although other hormones not measured here may as well be involved. We have previously shown that the aldosterone receptor regulates the expression of CD62L and CCR7[36], which induce adhesion of lymphocytes to the endothelium and are essential for the subsequent migration of the cells to lymph nodes[37,38]. Moreover, animal studies suggest that, during sleep, lymphocytes accumulate in lymph nodes[39,40]. Thus, the SO stimulation-induced decrease in circulating lymphocytes likely reflects a hormonally mediated redistribution of the cells to lymphoid tissues, where they can evolve their cell-specific immunological functions, eventually mediating the boosting effect of sleep on adaptive immune functions[3,14,36].

Effect sizes for lymphocyte numbers were surprisingly large, underlining the importance of SWS for immune functions. For the endocrine changes, effect sizes were medium. However, changes in cortisol were well comparable to previous experiments that manipulated SWS non-specifically using a pharmacological approach[22,41]. The similarity of the effects to these previous studies is indeed striking considering that, in the current experiment, we used a highly specific auditory stimulation that was subtle and lasted for only 120 min. Additionally, we stimulated healthy young men during the diurnal phase of maximum endogenous aldosterone and prolactin secretion, and minimal cortisol secretion. Hence, greater effects can be expected in conditions with acutely or chronically reduced amounts of SWS (e.g., after acute stress and during aging[42–44]), as observed in the above-mentioned pharmacological study, which indeed revealed larger effects in aged compared to young subjects[22]. In such conditions, enhanced SO activity after closed-loop stimulation might well compensate for respective changes in hormonal release during sleep, encompassing increased cortisol and reduced aldosterone and prolactin levels, which may contribute to the decline not only in immunity, but also in cognitive function in the elderly[21,45–48]. Moreover, effects of the stimulation might accumulate with repetitive application across several nights. Thus, besides advancing our understanding of the specific role of SWS in controlling peripheral body systems, our findings might have direct clinical relevance.

SO activity was a significant predictor of cortisol and aldosterone levels, despite controlling for the number of applied stimuli. Also, the number of auditory stimuli per se, although predicting SO activity, did not predict concentrations of the endocrine or immune parameters of interest. These findings point to a direct causal role of SOs in inducing the observed changes, and speak against the possibility that the auditory stimuli per se affected the endocrine/immune parameters independently of the increase in SO activity. Nevertheless, we cannot exclude that the auditory stimulation additionally affected other parameters that we did not measure here and that might have contributed to the observed endocrine and immunological changes. However, based on the literature showing clear associations between slow waves during sleep and the endocrine milieu, it is rather unlikely that the tones induced the observed changes through any unknown mechanism that is independent of the increase in SOs. Also, compared to other methods for manipulating SWS, like the administration of pharmacological agents, closed-loop stimulation is to our knowledge the most selective technique for inducing SOs and deepening SWS in humans so far, without affecting other sleep parameters. To test whether the effects are specific for the closed-loop stimulation, one might think of a control group stimulated in a random or open-loop fashion, i.e., by applying the same auditory stimuli during NonREM sleep independently of the endogenous brain rhythm. However, this approach is not necessarily effective: a previous study using this open-loop stimulation to test its effects on memory did not find a functional improvement in memory, but still an increased SO activity

(although less robustly) after open-loop stimulation[49]. In addition, this random application of stimuli during NonREM sleep changed other sleep parameters and was noticed during the night by half of the subjects, thus rendering such stimulation an inappropriate control[49]. In the current study, we therefore employed sham stimulation as the most "blank" control condition. Nevertheless, it remains a challenge to future studies to establish control conditions that enable to experimentally dissociate the effects of our closed-loop stimulation from those of the auditory stimulus presentation per se. This means, further efforts should be undertaken to establish a control condition of auditory stimulation that would not induce SOs or even robustly and selectively suppress them.

In sum, using closed-loop auditory stimulation to enhance sleep SOs, we provide first-time evidence suggesting a causal role for the sleep SOs in specifically regulating endocrine activity in support of T and B cell-mediated immunity. Closed-loop SO stimulation proved an effective tool for probing endocrine and immunological functions of SWS. It, thus, represents the first interventional approach that, beyond enhancing phenotypic sleep, concurrently improves its endocrine-regulatory and immune-supportive function. These actions make closed-loop SO stimulation a promising clinical approach that, as an easy-to-use and highly specific technique, might eventually replace traditional pharmacological treatments of insomnia and depression, i.e., diseases presenting not only with disturbed sleep, but also with hormonal and immunological abnormalities[50–52].

## Methods

**Participants and experimental procedure.** Fourteen physically and mentally healthy men (mean age 24 years $\pm 2.16$ s.d., mean BMI 23 kg/m$^2$ $\pm$ 2.14 s.d.) participated in this randomized, within-subject cross-over study. Each of two experimental conditions started at 2100 hours with preparing polysomnography and insertion of a catheter for blood sampling (~1.5 h before the first blood sampling). The subject was then allowed to sleep from 2300 to 0700 hours. In the Stimulation condition, two low-intensity tones were delivered (by in-ear headphones) whenever an endogenous SO was identified electroencephalographically during NonREM sleep. Stimulation started with the onset of consolidated NonREM sleep (i.e., 10 min after visual detection of five slow waves with an amplitude >80 µV within 30 s in the EEG), and terminated after 120 min independently of the current sleep stage. In the Sham condition, only time points of potential stimulation were marked, but no stimulation was applied. Blood for assessment of hormone concentration was sampled via an intravenous catheter connected to a long thin tube that enabled blood collection from an adjacent room without disturbing the subject's sleep. Samples were drawn at 2245 hours, 2300 hours, immediately before the start of stimulation (baseline values), then every 5 min during stimulation, and (in $n = 11$) every 15 min following the end of stimulation until 0700 hours. T and B lymphocyte counts were assessed in blood samples drawn at baseline, 1, 2, 3, and 4 h after stimulation onset, and after awakening at 0700 hours ($n = 9$). The two conditions were separated by at least 2 weeks (and not more than 4 weeks) and randomization was performed in a semi-automated manner, ensuring that the order of conditions was balanced across subjects. Subjects were blind to the condition, which was overall confirmed by the fact that only two out of 14 subjects reported having shortly noticed auditory stimuli at the beginning of the night. Analyses of EEG data, hormones, and immune parameters were performed with the experimenter being unaware of the condition.

All participants had a regular sleep/wake rhythm, did not present any sleep disturbances, were not taking any medication at the time of the experiments, and were non-smokers. Acute and chronic illness was excluded by medical history, physical examination, and routine laboratory investigation. The men were synchronized by daily activities and nocturnal rest. All subjects spent one adaptation night in the laboratory in order to habituate to the experimental setting. The study was approved by the Ethics Committee of the University of Tübingen, and all participants gave written informed consent.

**Sleep recordings and auditory stimulation.** The EEG was recorded continuously during sleep with a BrainAmp DC amplifier (Brain Products GmbH, Gilching, Germany) from nine locations (international 10–20 system; F3, Fz, F4, C3, Cz, C4, P3, Pz, and P4) referenced to the average potential from electrodes attached to the mastoids (M1 and M2). A ground electrode was attached to the forehead. Ag–AgCl electrodes were used, and impedances were always kept below 5 kΩ. Signals were filtered between 0.03 and 250 Hz, sampled at 500 Hz, and stored for later offline analysis on a PC together with the stimulation triggers. Additionally, eye movements (EOG), the electromyogram (EMG) from the chin and electrocardiogram

(ECG) were recorded for standard polysomnography. Sleep stages were determined off line using EEG recordings from C3, C4, EOG, and EMG for subsequent 30 s epochs according to standard criteria[53]. Scoring according to these criteria of Rechtschaffen and Kales allows a fine-grained analysis of SWS by discriminating between sleep stages S3 and S4, for which normative data exists[54].

A separate EEG recording system was used to accomplish the online detection of SOs and present the auditory stimuli. This system consisted of a "Digitimer D360" EEG amplifier (Digitimer LTD, Hertfortshire, UK) and a "Power1401 mk 2" high-performance data acquisition interface (Cambridge Electronic Design, Cambridge, UK) connected to a separate PC. With this setup, the EEG was recorded from an electrode positioned at AFz, with reference to the average potential from linked electrodes attached to the earlobes. The EEG was filtered between 0.25 and 4 Hz and sampled with 200 Hz. The auditory stimulation was triggered when a negative SO peak was about to occur, i.e., each time the filtered EEG signal crossed an adaptive threshold toward larger negative values. On default, this threshold was set to $-80 \mu V$. Every 0.5 s, it was updated to the minimal (i.e., largest negative) instantaneous EEG amplitude within the preceding 5-s interval, however, only if this value exceeded (in negativity) $-80 \mu V$. For each subject, the SO detection algorithm was also applied (offline) to the first SWS epoch of the adaptation night, in order to determine the subject's individual delay time between the detected negative half-wave peak and the succeeding depolarizing upstate, i.e., the mean time between the SO-negative peak and the following positive peak. This time was used to individually adapt the stimulation in the experimental nights such that the auditory stimuli were most likely to occur in phase with the SO upstate. Upon detection of a SO-negative peak, the first stimulus was delivered after this individually adjusted delay time; the second stimulus then followed after a fixed interval of 1075 ms. Then, stimulation was discontinued for 2.5 s, to mimic the typical temporal organization of spontaneous SOs occurring in trains of two or three succeeding waves[55], before the detection algorithm started again. The detection algorithm was applied throughout the stimulation period with a total duration of 120 min (starting with the onset of the first consolidated NonREM sleep epoch), but halted whenever the subject left NonREM sleep stage 2 or SWS or arousals occurred. During the Sham condition, SO detection was performed in the same way, and the respective time points were marked in the EEG, but no auditory stimuli were delivered. The auditory stimuli were presented binaurally via MDR-EX50LP in-ear headphones (Sony Deutschland). They consisted of bursts of pink 1/f noise of 50 ms duration with a 5 ms rising and falling time, respectively. Sound volume was calibrated to 50 dB SPL. Responsiveness of subjects to the stimulation was confirmed by measuring the individual's averaged evoked potential response to the tones.

**Spectral analysis and offline analysis of SOs.** Power spectra were determined by Fast Fourier Transformation (FFT) using a Hanning window with 4096 data points (~8.2 s), resulting in a frequency resolution of 0.122 Hz. Subsequent windows overlapped by 50%. The power spectra were averaged across all 8.2-s windows and subsequently smoothed with a three-point moving average. To account for individual variability, we normalized the power spectrum for each subject by dividing it by its cumulative power up to 30 Hz. The unit of the cumulative power, i.e., the area under the curve, corresponds to $\mu V^2$ x Hz, hence the normalization yields the unit $Hz^{-1}$.

Offline detection of SOs was based on an established algorithm[56]: This algorithm is based on a virtual channel representing the mean EEG signal recorded from F3, Fz, F4, C3, Cz, C4, P3, Pz, and P4 and was applied to SWS epochs across the entire night. In brief, the EEG is first low-pass filtered at 30 Hz and down sampled to 100 Hz. For the identification of large SOs, a low-pass filter of 3.5 Hz is applied. Then, mean negative and positive peak potentials are derived from all intervals between consecutive positive-to-negative zero crossings with a duration between 0.833 and 2 s (corresponding to a frequency of 0.5–1.2 Hz). We then marked those intervals as SO cycles where the negative peak amplitude was lower than 1.25 times the mean negative peak amplitude and where the amplitude difference (positive peak minus negative peak) was larger than 1.25 times the mean amplitude difference. Negative half-wave peaks were used to mark SO events.

**Hormone and lymphocyte analyses.** Cortisol, prolactin, GH, and ACTH concentrations were assessed in plasma using commercial assays (Siemens Healthcare Diagnostics Inc., Tarrytown, NY, USA) and aldosterone levels were measured in plasma by ELISA (DRG Instruments GmbH, Marburg, Germany). Sensitivity and intra-assay and interassay variability were as follows: Cortisol: 5.5 nmol/l, <5.5%; prolactin: 0.3 µg/l, <4.9%; GH: 0.05 µg/l, <6.6%; ACTH: 1.1 nmol/l, <10.0%; aldosterone: 5.7 ng/l, <9.4%. ACTH levels were mostly below the detection threshold during the stimulation period, and are therefore not reported.

Absolute counts of T and B lymphocytes were determined by a "lyse no-wash" flow cytometry procedure. Briefly, 50 µl of an undiluted blood sample was immunostained with anti-CD45 (clone HI30), anti-CD3 (clone UCHT1), and anti-CD19 (clone SJ25C1) antibodies in Trucount tubes (BD Biosciences, San Jose, CA, USA). After 15 min of incubation at room temperature, 0.9 ml of FACS lysing solution (BD Biosciences) was added to lyse erythrocytes for 15 min. Finally,

samples were mixed gently, and at least 100,000 CD45$^+$ cells were acquired on a BD LSRFortessa Flow Cytometer (BD Biosciences).

**Statistical analyses.** Differences between conditions were analyzed with two-sided Student's t tests for sleep and EEG data and with two-sided Wilcoxon-signed-rank tests for endocrine and immune parameters as these data were not normally distributed. To reduce type 1 error with multiple comparisons of time series, we clustered hormonal data into 1-h bins for statistical testing. A p value <0.05 was considered statistically significant. We calculated the effect size Cohen's d for the impact of the stimulation on SO activity and on the amplitude of SOs, and followed Cohen's criteria for interpretation of the sizes ($d = 0.2$, small; $d = 0.5$, medium; $d = 0.8$, large)[57]. For the impact of SO stimulation on endocrine and immune parameters, we calculated the effect size estimate r, which is used instead of Cohen's d for non-parametric tests, with the following criteria for interpretation of the sizes: $r = 0.1$, small; $r = 0.3$, medium; $r = 0.5$, large[57].

Correlations of mean SO activity during the stimulation interval with endocrine/immune parameters during time intervals of significant effects and with the number of applied auditory stimuli were calculated using Spearman's rho. The correlations remained non-significant ($r < 0.3$, $p > 0.289$), presumably due to the low between-subject variance in SO activity during the stimulation interval (see ref. [12] for a comparable lack of correlation) and were not reported here in detail. Hence, at a second step, we performed hierarchical linear regression analyses including the parameters of interest over an extended period, i.e., the first four 1-h bins post-stimulation onset, which is the time with predominant SWS. These analyses included SO activity as predictor variable and the different hormone/lymphocytes measures as dependent variables, while correcting for the factor "Time bin" (to control for variance explained by inclusion of the four time bins per subject). To control for possible contributions of the auditory stimuli per se, the analyses were additionally corrected for the factor "Number of applied auditory stimuli." Further analyses were performed with the number of auditory stimuli as predictor variable for SO activity and for endocrine/immune parameters. A distribution-independent bootstrapping procedure with 10,000 samples was used for the regressions because endocrine/immune parameters and the number of applied stimuli were not normally distributed. To account for delayed effects of the stimulation on aldosterone and lymphocytes, we performed the analyses including a delay (time lag) of the SO activity of 0, 1, 2, and 3 h relative to these peripheral parameters.

The sample size was chosen based on previous studies that manipulated SWS non-specifically using pharmacological agents[22,41]. For the first three subjects, we started with a blood sampling rate of 2.5 min for the stimulation interval, which precluded collecting more blood post stimulation due to ethical limitations in the amount of blood that can be collected per subject. For the other 11 subjects, we reduced the blood sampling rate during the stimulation interval to 5 min, as this was still high enough to detect even rapid changes in endocrine levels and allowed us to collect blood also post stimulation. One subject had to be excluded post hoc from the analysis of aldosterone concentrations, as he showed unusually high levels of this hormone in both conditions. Immune parameters were collected in a subset of nine subjects.

**Data availability.** Data supporting the findings of this study as well as the computer code for the algorithm for SO detection are available from the corresponding author on reasonable request.

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

## Acknowledgements

We are grateful to Ilja Goldins and Samuel Büttner for technical assistance. This work was supported by grants from the Deutsche Forschungsgemeinschaft (TR-SFB 654) and

from the German Federal Ministry of Education and Research to the German Center for Diabetes Research (01GI0925).

## Author contributions

L.B., H.-V.V.N. and J.B. conceived and designed the study; L.B., H.-V.V.N. and C.G. performed the experiments; L.B., H.-V.V.N. and R.L. analyzed the data; S.D. helped with study design and data analysis; L.B. drafted the manuscript; H.-V.V.N., S.D., R.L., C.G. and J.B. revised and approved the final version of the manuscript.

## Additional information

**Competing interests:** The authors declare no competing financial interests.

