## [Peer Review File · Nature Communications]

Reviewers' expertise:

Reviewer #1: Sleep, EEG analysis;
Reviewer #2: Sleep, EEG analysis;
Reviewer #3: Sleep, endocrinology.

Reviewers' comments:

Reviewer #1 (Remarks to the Author):

This study investigates the effects of auditory stimulation which leads to enhancement of slow oscillations (SOs) during sleep, on endocrine and immune variables. The results are interesting. These are my concerns

1. It is claimed that the study shows a causal role for SOs in regulating cortisol etc [Page 6, line 105]. This is not true. The study only shows that auditory stimulation (which happens to be associated with changes in SO's) induces some endocrine changes. For the claim to be justified the control condition should be administration of auditory stimuli which do not induce SOs (or reduce SOs). Only if the endocrine responses differ between the active and these control conditions (or control condition) is the conclusion justified.
2. It is unclear whether the auditory stimulation exclusively enhanced SO's or also sleep spindles etc. EEG Power spectra during stimulation need to be shown
3. Please clarify whether SWS during stimulation was enhanced. Table 1 only shows all night values.
4. The statistical analysis appears to be somewhat 'pragmatic'. Although the cortisol time course is presented and is somewhat convincing [Although it is unclear why the SEMs for the sham are so much larger than for the active condition, and this may indicate that the data are not normally distributed] for the other hormones time courses are not presented, and only averages over the intervals for which the time point contrast were significant (Fig 2C,D). It seems that the intervals shown were based on the post-hoc inspection of the results. In general, the data should be analysed with a standard mixed model ANOVA with repeated measures.
5. Details of the protocol are unclear.
 - 5a) What was the washout period between the two conditions?
 - 5b) Was the order of 'control' and 'sham' balanced?
 - 5c) Was the 'second condition' preceded by an adaptation night?
 - 5d) if 5c is not then the statistical model should have 'period (1, or 2) as a separate factor
 - 5e: Methods to reduce bias: It needs to be stated whether all data were analysed blind to condition or not.
6. The authors need to distinguish between effects of SOs and SWS, or at least clarify how this terminology is used
7. The authors may consider mentioning the effects (or absence thereof) of SWS reduction (through acoustic stimulation) on insulin sensitivity measures (Tasali et al. PMID: 18172212) or body temperature measures (Beersma, PMID: 10607052) as examples of divergent effects of manipulation of SWS.

Reviewer #2 (Remarks to the Author):

Very few studies have used closed-loop acoustic stimulation to deepen sleep. First papers were presented by the group of Jan Born in 2013. In this manuscript the same method is applied in young healthy adults to investigate the effects of boosting slow wave sleep during the initial period of sleep on several immunological and endocrinological markers. Thus this manuscripts combines a very promising novel technology and a research area "sleep and health" which is only marginally explored in the human literature, particularly, when it comes to investigating causal relationships. For both aspects this is an important contribution and should interest a wide audience. However, there are some limitations which should be addressed:

Statistics:

No ANOVAs were applied. This is essential for data sets where multiple time points/factors are investigated (e.g. Figure 2a, d, e). Here, e.g., factors "condition" and "time" need to be included. Also there is no correction for multiple testing applied. This correction also needs to include tests which are not presented here but actually tested (e.g. other time points). Finally, one-tailed tests were used which should be more clearly stated in the manuscript (not only in the Methods).

Blinding:

Were subjects blind to the condition? Was it verified, e.g. by asking in the morning, whether they were aware of the acoustic stimulation. This is important because the expectation of getting a "sleep support" may already affect the investigated endocrine markers. The randomization procedure should be mentioned.

Also related to the data analysis blinding is of importance. For example, were experimenters performing the sleep scoring blind to the condition?

Specificity of the stimulation:

The authors write that their stimulation "distinctly increases slow oscillatory EEG activity". However, the effects of the stimulation on other frequencies is not presented. Ideally a power spectrum for the duration of stimulation is presented for both conditions and compared statistically.

Moreover, slow wave activity (SWA) is strongly regulated. Thus, the question arises whether deepening sleep in the first period of sleep has an impact on subsequent sleep (e.g. second half of the night). Based on current models maybe decreased SWA during later periods might be expected. To address this point an analysis of the time course of SWA across the night would be needed. This is of particular importance since some of the effects are present after the stimulation.

Direct relationship:

Individual differences in the stimulation effect on SWA as well as its effect on endocrine markers exists. It would be interesting to test whether a relationship between these factors exists (or not). Thus correlation analysis should be presented.

Minor points:

Expressions related to the quantification of slow waves might be confusing: Slow wave activity, SO activity, slow waves sleep is used. When is what adequate and what is the difference?

The authors write "stimulation started with the onset of consolidated NonREM sleep". How was this defined? Was this variable across subjects? Please provide a stimulation onset parameter. Moreover, stimulation was "terminated after 120 min". Was this independent of the sleep structure of a subject, i.e. including waking and REM sleep? As a consequence the number of acoustic stimulation may vary a lot across subject. Please provide this data.

Subject numbers vary for the different parameters. Please explain carefully.

In the discussion about prolactin an endogenous rhythm in the slow wave frequency range is mentioned. Could that mean that the effects were not driven by changes in SO activity but rather direct effects occurred? If no correlation between SO activity and endocrine markers exists that might be an interpretation which should be discussed.

Why was scoring not done according to the novel criteria of the AASM Manual? Including frontal channels in the scoring (as proposed in the manual) may reveal more SWS (under stimulation).

Why was the "stimulation discontinued for 2.5 s"?

Reviewer #3 (Remarks to the Author):

This paper addresses a very important and novel topic likely to be of interest to a wide audience. The work reported directly addresses the question of whether a non-invasive non-pharmacological method that can increase the amplitude of cortical slow oscillations also stimulates extra-cortical brain mechanisms engaged by physiological SWS and the peripheral benefits of physiological SWS. To address this issue, the authors have completed a randomized controlled cross-over trial (effective acoustic stimulation versus a sham condition) in 14 healthy young men and obtained blood samples at frequent intervals throughout the night. GH, prolactin, cortisol and aldosterone were assayed on each sample. These hormones exert modulatory effects on immune function and therefore T and B cell counts were also measured.

MAJOR COMMENTS

1. The lack of stimulation of GH is a major negative finding that needs to be mentioned in the abstract and discussed further. The stimulation of GH release by SWS, particularly in healthy young men who were the participants in the present study, is the best-documented mechanistic connection between SWS and peripheral function. Further, there is abundant

experimental evidence for a dose-response relationship between the amount of SWA and the increase in plasma GH levels. In contrast, the release of other hormones, including prolactin and aldosterone, and the inhibition of ACTH and cortisol, are temporally coincident with episodes of SWS, but a dose-response relationship has not been observed.

2. An increase in prolactin release is not seen "concomitantly to deepening SWS" (contrary to the statement in the Abstract) but rather occurs during the hour following the end of stimulation. Can you exclude a rebound effect where, when acoustic stimuli stop, dopaminergic activity is better suppressed and prolactin increases ?

3. An increase in aldosterone is not seen "concomitantly to deepening SWS" (contrary to the statement in the Abstract) but rather occurs 3-6 hours after the stimulation has ended.

4. A reduction of cortisol levels was indeed observed during the first hour of stimulation in the active versus sham condition. However, the SEM for the mean cortisol levels in the sham condition (Fig. 2a) is 3-4 times larger than for the active condition, which could reflect outliers. Have you examined a possible order effect of the two conditions ? When was the sampling catheter inserted relative to the beginning of sampling ?

5. The BMI of the subjects should be reported. Adiposity suppresses GH release and this could be an explanation for the lack of effect on GH.

SUGGESTIONS

1. A figure showing the entire period of blood sampling, concomitantly with the profiles of SO activity, should be used to report mean profiles of GH, cortisol, prolactin and aldosterone levels as well B/T cell counts in each condition.

2. In Figure 1, it is unclear whether tones were delivered when SO were present in NREM sleep (including N2). Panel 1c reports mean SO activity and amplitude for epochs of SWS (N3, N2 excluded). The mean number of SWS epochs used in the calculation of the data shown in 1c should be reported in the legend for each condition.

3. The statistical analysis needs to be repeated using two-tail tests since the result opposite to the hypothesis (for example: acoustic stimulation stimulates sympathetic nervous activity, preventing concomitant GH & prolactin release ??) cannot be excluded.

4. Differences in temporal profiles between conditions (active versus sham) should be analyzed by ANOVA for repeated measures with condition as factor, time as repeated measure and the interaction time x condition.

In detail, the following changes have been introduced regarding the referees' comments (highlighted in yellow):

Reviewer #1 (Remarks to the Author):

This study investigates the effects of auditory stimulation which leads to enhancement of slow oscillations (SOs) during sleep, on endocrine and immune variables. The results are interesting. These are my concerns

1. It is claimed that the study shows a causal role for SOs in regulating cortisol etc [Page 6, line 105]. This is not true. The study only shows that auditory stimulation (which happens to be associated with changes in SO's) induces some endocrine changes. For the claim to be justified the control condition should be administration of auditory stimuli which do not induce SOs (or reduce SOs). Only if the endocrine responses differ between the active and these control conditions (or control condition) is the conclusion justified.

Authors' response:

Thank you very much for raising this important issue. Indeed, we cannot entirely rule out that the observed endocrine changes were not directly caused by the SOs. To our knowledge, the applied closed-loop stimulation is so far the most selective method for specifically enhancing SOs without changing the general sleep architecture or spectral power in other frequencies (in contrast to an open-loop approach, see e.g., Weigenand et al., 2016. *Eur J Neurosci*). We principally cannot rule out that closed-loop stimulation induced the endocrine changes independently of the increase in SO activity, via any other parameters not measured here. Importantly, including a control condition with auditory stimuli that do not induce SOs would still not definitely prove that the SOs are the direct cause of the endocrine changes, because this type of stimulation might also leave a third parameter unaffected that could have potentially caused the endocrine changes. The issue of finding the perfect control is further complicated by the fact that applying the same auditory stimulation in a random way does also induce an auditory event related potential and can therefore also enhance SO activity, although in a less specific way (Weigenand et al., 2016. *Eur J Neurosci*). This is the main reason why we decided to include sham stimulation as the most "blank" control. However, we acknowledge that attributing a direct causal role to SOs in inducing the hormonal changes is not correct in a strict sense. We have therefore now moderated our statement on the causality, and discussed the possibility that other factors might also be involved. Nevertheless, this is the first study that modulates the endocrine milieu by enhancing SOs in the most specific way possible, and it shows that with this stimulation method one can intensify the endocrine and immunological profile in a direction that is consistent with changes observed during physiological sleep. Of course, to more precisely characterize the underlying mechanisms, further studies need to be done (including control groups with different types of stimulation). We adapted the manuscript as follows:

Discussion, page 6: "Considering that our method is highly specific in selectively enhancing SOs without affecting other sleep parameters, our findings suggest a causal role of SOs in inducing the hormonal milieu characterizing SWS.

Discussion, pages 9f: "We cannot exclude that the auditory stimulation also affected other parameters that we did not measure here and that might have contributed to the observed endocrine and immunological changes. However, based on the literature showing clear associations between slow waves during sleep and the endocrine milieu, it is rather unlikely that the tones induced the observed changes through any unknown mechanism that is independent of the increase

in SOs. Also, compared to other methods for manipulating SWS, like the administration of pharmacological agents, closed-loop stimulation is to our knowledge the most selective technique for inducing SOs and deepening SWS in humans so far, without affecting other sleep parameters. To test whether the effects are specific for the closed-loop stimulation one might think of a control group stimulated in a random or open-loop fashion, i.e., by applying the same auditory stimuli during NonREM sleep independently of the endogenous brain rhythm. However, a previous study using this open-loop stimulation to test its effects on memory did not find a functional improvement in memory, but still an increased SO activity (although less robustly) after open-loop stimulation, thus rendering such a stimulation an inappropriate control⁴⁹. Nevertheless, further studies are needed to establish whether the endocrine and immunological changes that we observed here depend (like the effects on memory) specifically on the closed-loop approach or can also be induced by random application of tones that induce slow oscillatory responses. ... In sum, using closed-loop auditory stimulation to enhance sleep SOs, we provide first-time evidence suggesting a causal role for the sleep SOs in specifically regulating endocrine activity in support of T and B cell-mediated immunity."

2. It is unclear whether the auditory stimulation exclusively enhanced SO's or also sleep spindles etc. EEG Power spectra during stimulation need to be shown

Authors' response:

We have now calculated EEG power spectra for the stimulation interval. The auditory stimulation did only affect frequencies in the slow oscillation range. Spindles and other frequencies were not changed. We have now added this information to our manuscript as follows:

Results, page 5: "...as well as the amplitude of SOs ($p = 0.003$ and $p = 0.024$, respectively, for electrode position Fz; Fig. 1b,c). Other frequencies remained unchanged and SO activity was only affected during the stimulation interval but not afterwards (Supplementary Figs. 1, 2)."

Supplementary Figure 1: Auditory closed-loop stimulation selectively enhances slow oscillatory activity. Mean (\pm s.e.m.) spectral power during stimulation for electrode position Fz determined for NonREM epochs of the 120-min stimulation period for the Stimulation (black) and Sham condition (grey) for frequencies up to 30 Hz. The bottom panel indicates significance between the effects of the Stimulation and Sham condition (paired t-tests, two-sided). The insert shows the power in the SO peak frequency band (0.8-1.1 Hz); $**p < 0.01$. $n = 14$.

3. Please clarify whether SWS during stimulation was enhanced. Table 1 only shows all night values.

Authors' response:

SWS duration in minutes or as a percent of total sleep time was unchanged during as well as post stimulation. We now included this information in the manuscript:

Results, page 5: "The absolute and percent time spent in the different sleep stages remained unaffected by the stimulation (Table 1), with this pattern overall confirming the specificity of this method selectively enhancing SOs. SWS duration in minutes or as a percent of total sleep time was also not affected in separate analyses of the stimulation interval and the post-stimulation interval ($27\% \pm 3.3$ s.e.m. for the Stimulation condition and $30\% \pm 4.2$ s.e.m. for the Sham condition, $p > 0.351$)."

4. The statistical analysis appears to be somewhat 'pragmatic'. Although the cortisol time course is presented and is somewhat convincing [Although it is unclear why the SEMs for the sham are so much larger than for the active condition, and this may indicate that the data are not normally distributed] for the other hormones time courses are not presented, and only averages over the intervals for which the time point contrast were significant (Fig 2C,D). It seems that the intervals shown were based on the post-hoc inspection of the results. In general, the data should be analysed with a standard mixed model ANOVA with repeated measures.

Authors' response:

We thank the reviewer for raising this important point. Please note for several time points, the hormonal and immune data were not normally distributed (even after logarithmic or square root transformation) which precludes straightforward application of ANOVA. We therefore have now employed non-parametric statistical analyses (Wilcoxon tests). To reduce the inflation of the Type I error, we clustered the data and analyzed them only for one-hour bins, which comes closer to running an ANOVA including the Factors "Condition" and "Time" than our previous approach (please, also refer to our answer to comment #1 by reviewer 2). Using this statistical approach, the results remain essentially the same as those previously reported. The one-hour bins are now based on the time (1, 2, 3, 4, 5, 6, 7, and 8 hours post stimulation onset), instead of a post-hoc clustering of those adjacent time points that show significance between conditions (see new Supplementary Fig. 3). We adapted the manuscript accordingly:

Results, page 5: "Stimulation distinctly reduced blood cortisol concentrations during the first hour after stimulation onset ($p = 0.024$, Fig. 2a,b, Supplementary Fig. 3a), although levels were already rather low during this early-night interval. The reduction was visible already 5 min post stimulation onset and averaged 15% in the first hour of stimulation (Fig. 2a). Prolactin levels were increased in the Stimulation compared to the Sham condition during the 3rd hour after stimulation onset ($p = 0.038$; Fig. 2c, Supplementary Fig. 3b). Levels of aldosterone were increased during the 4th and 8th hour after stimulation onset ($p = 0.014$ and 0.037 , respectively; Fig. 2d, Supplementary Fig. 3c). GH remained unchanged (Supplementary Fig. 3d). Effect sizes (r) were -0.37 for cortisol, 0.38 for prolactin, and 0.49 and 0.40 respectively, for aldosterone, which reflect medium-sized effects. Blood T and B lymphocyte counts were acutely reduced 3 hours post stimulation onset ($p = 0.006$ and $p = 0.011$, respectively; Fig. 3). Respective effect sizes were -0.60 for T cells and -0.54 for B cells, which are considered large effect sizes."

Methods, page 15: “Differences between conditions were analyzed with Student’s t-tests for sleep and EEG data and with Wilcoxon-signed-rank tests for endocrine and immune parameters as these data were not normally distributed. Tests were one or two-sided, respectively, for directed vs. undirected hypothesis testing. To reduce Type 1 error with multiple comparisons of time series we clustered hormonal data into one-hour bins for statistical testing. A p-value < 0.05 was considered statistically significant. We calculated the effect size estimate r for the impact of SO stimulation on endocrine and immune parameters and followed Cohen’s criteria for interpretation of the sizes ($r = 0.1$, small; $r = 0.3$, medium; $r = 0.5$, large)⁵⁷. Correlations between the increase in SO activity during the stimulation interval and changes in endocrine and immunological parameters at the time points of significant effects were calculated with Spearman’s rho. The sample size was chosen based on previous studies that manipulated SWS non-specifically using pharmacological agents^{22,41}.”

Figure 2. Impact of auditory SO stimulation on cortisol, prolactin, and aldosterone levels. Means (\pm s.e.m.) of cortisol (a,b), prolactin (c) and aldosterone (d) levels calculated for one-hour bins. Grey area represents the 120-min stimulation period. * $p < 0.05$ for pairwise comparisons between the Stimulation condition (STIM, black) and the Sham condition (SHAM, white) with Wilcoxon tests, one sided. $n = 10-14$ (see Methods section for exact numbers).

Figure 3. Impact of auditory SO stimulation on lymphocyte counts. Means (\pm s.e.m.) of circulating T and B cell numbers (shown as difference from baseline). Grey area represents the 120-min stimulation period. * $p < 0.05$, ** $p < 0.01$ for pairwise comparisons between the Stimulation condition (STIM, black) and the Sham condition (SHAM, white) with Wilcoxon tests, one-sided. $n = 9$.

Supplementary Figure 3. Time course of cortisol, prolactin, aldosterone and growth hormone levels with and without auditory SO stimulation. Means (\pm s.e.m.) of cortisol (a), prolactin (b), aldosterone (c), and growth hormone (d) levels calculated for one-hour bins starting from stimulation onset. Grey area represents the 120-min stimulation period. * $p < 0.05$ for pairwise comparisons between the Stimulation condition (STIM, black) and the Sham condition (SHAM, white) with Wilcoxon tests, one-sided. $n = 10-14$ (see Methods section for exact numbers). Please note in (a) the different scaling of the y-axis for cortisol levels during and post stimulation, respectively, which was applied because of the strong circadian rhythm of this hormone.

5. *Details of the protocol are unclear.*

5a) *What was the washout period between the two conditions?*

5b) *Was the order of 'control' and 'sham' balanced?*

5c) *Was the 'second condition' preceded by an adaptation night?*

5d) *if 5 c is not then the statistical model should have 'period (1, or 2) as a separate factor*

5e: *Methods to reduce bias: It needs to be stated whether all data were analysed blind to condition or not.*

Authors' response:

We now describe the protocol in more detail. We included the information that the “washout” period between the two conditions was at least two weeks (14-26 days), that all data were analyzed blind to the condition and that the order of conditions was balanced. There was only one adaptation night, which took place several days before the first condition. Available literature suggests that there is no need of including a second adaptation night before the second experimental night as there is no difference in sleep parameters between the first experimental night (i.e., the second night in the sleep lab including the adaptation night) and succeeding ones (e.g., Lorenzo & Barbanoj, 2002. *Psychophysiology*). In addition, in our study, the adaptation night preceded the first experimental night on average by a similar number of days as the first experimental night preceded the second experimental night (13.6 days versus 18.7 days; not significantly different). Therefore, the first experimental night can be regarded as an adaptation night for the second experimental night. The balanced order of conditions additionally prevented a systematic influence of the order of conditions. Finally, we also ran some exploratory analyses using the factor “order” (first stimulation vs first sham condition), which did not reveal any significant effects of this factor on the target variables.

Methods, page 11: “The two conditions were separated by at least two weeks (and not more than four weeks) and randomization was performed in a semi-automated manner, ensuring that the order of conditions was balanced across subjects. Subjects were blind to the condition, which was overall confirmed by the fact that only two out of 14 subjects reported having shortly noticed auditory stimuli at the beginning of the night. Analyses of EEG data, hormones and immune parameters were performed with the experimenter being unaware of the condition.

All participants had a regular sleep/wake rhythm, ...”

6. *The authors need to distinguish between effects of SOs and SWS, or at least clarify how this terminology is used*

Authors' response:

We used the term slow wave activity (SWA) to refer to the frequency range between 0.5 and 4 Hz, whereas SO activity includes only frequencies ≤ 1 Hz. Since we specifically enhanced the SO frequency and there was no change in the frequency range covering all slow waves (i.e. 0.5 – 4 Hz), we assume that the effects are mediated specifically by SO activity and not by slow waves in general. When referring to the results of the current study, we therefore used the term SO activity. When citing studies that did not differentiate between SO activity and SWA, we used the term SWA. SWS refers to the deepest sleep stage, which is characterized by slow waves in the EEG, including the SOs. We now explained the terms more clearly and used this terminology in a stricter manner:

Abstract: “Here, we enhance SOs in men by closed-loop stimulation, i.e., by delivering tones in synchrony with endogenous SOs.”

Introduction, page 3: “SWS is hallmarked by slow waves in the electroencephalogram (EEG), which have a frequency of 0.5-4 Hz and include the slow oscillation (SO) frequencies ≤ 1 Hz. Slow wave activity (i.e., the spectral power in the frequency range of 0.5-4 Hz) is associated with the co-ordinate release of immune-active hormones, specifically with a suppression of cortisol and an increase in prolactin, growth hormone (GH) and aldosterone levels, which provides an optimal endocrine milieu for supporting adaptive immune functions^{3,5-7}. ... However, so far, evidence for the relationship between slow wave or SO activity and the endocrine and immune functions of interest is solely based on correlational findings and, to the best of our knowledge, there is no direct proof of a causal role of SOs and SWS in this relationship. ... However, there have been no attempts to specifically strengthen SWS or SOs to boost its accompanying endocrine and immunological changes.”

Discussion, page 9: “In such conditions, enhanced SO activity after closed-loop stimulation might well compensate for respective changes in hormonal release during sleep, ...”

7. The authors may consider mentioning the effects (or absence thereof) of SWS reduction (through acoustic stimulation) on insulin sensitivity measures (Tasali et al. PMID: 18172212) or body temperature measures (Beersma, PMID: 10607052) as examples of divergent effects of manipulation of SWS.

Authors' response:

We now mention the publication by Tasali et al. and other studies as examples for studies using acoustic stimulation to suppress SWS.

Introduction, page 3: “However, so far, evidence for the relationship between slow wave or SO activity and the endocrine and immune functions of interest is solely based on correlational findings and, to the best of our knowledge, there is no direct proof of a causal role of SOs and SWS in this relationship. There are a few studies showing that selective suppression of SWS can affect peripheral functions, like glucose tolerance and blood pressure⁹⁻¹¹. However, there have been no attempts to specifically strengthen SWS or SOs to boost its accompanying endocrine and immunological changes.”

Reviewer #2 (Remarks to the Author):

Very few studies have used closed-loop acoustic stimulation to deepen sleep. First papers were presented by the group of Jan Born in 2013. In this manuscript the same method is applied in young healthy adults to investigate the effects of boosting slow wave sleep during the initial period of sleep on several immunological and endocrinological markers. Thus this manuscript combines a very promising novel technology and a research area “sleep and health” which is only marginally explored in the human literature, particularly, when it comes to investigating causal relationships. For both aspects this is an important contribution and should interest a wide audience. However, there are some limitations which should be addressed:

1. Statistics:

No ANOVAs were applied. This is essential for data sets where multiple time points/factors are investigated (e.g. Figure 2a, d, e). Here, e.g., factors “condition” and “time” need to be included. Also there is no correction for multiple testing applied. This correction also needs to include tests which are not presented here but actually tested (e.g. other time points). Finally, one-tailed tests were used which should be more clearly stated in the manuscript (not only in the Methods).

Authors’ response:

We thank the reviewer for raising this important point. Please note for several time points, the hormonal and immune data were not normally distributed (even after logarithmic or square root transformation; please refer also to comment #4 by reviewer 1) which precludes straightforward application of ANOVA. We therefore have now employed non-parametric statistical analyses (Wilcoxon tests). To reduce the inflation of the Type I error, we clustered the data and analyzed them only for one-hour bins, which comes closer to running an ANOVA including the Factors “Condition” and “Time” than our previous approach. Using this statistical approach, the results remain essentially the same as those previously reported. The results would not survive a Bonferroni correction, but this approach would be too conservative as the different values are highly interdependent, and would strongly increase the Type II error (see also Rothman, 1990. *Epidemiology*). The fact that 5 out of 6 parameters (cortisol, aldosterone, prolactin, growth hormone, T cells, B cells) were changed in the expected direction and with moderate to large effect sizes, additionally speaks against the possibility that our results were observed only by chance. We adapted the manuscript accordingly and additionally now also state in the Figure legends whether tests were one or two-tailed:

Results, pages 5: “Stimulation distinctly reduced blood cortisol concentrations during the first hour after stimulation onset ($p = 0.024$, Fig. 2a,b, Supplementary Fig. 3a), although levels were already rather low during this early-night interval. The reduction was visible already 5 min post stimulation onset and averaged 15% in the first hour of stimulation (Fig. 2a). Prolactin levels were increased in the Stimulation compared to the Sham condition during the 3rd hour after stimulation onset ($p = 0.038$; Fig. 2c, Supplementary Fig. 3b). Levels of aldosterone were increased during the 4th and 8th hour after stimulation onset ($p = 0.014$ and 0.037 , respectively; Fig. 2d, Supplementary Fig. 3c). GH remained unchanged (Supplementary Fig. 3d). Effect sizes (r) were -0.37 for cortisol, 0.38 for prolactin, and 0.49 and 0.40 respectively, for aldosterone, which reflect medium-sized effects. Blood T and B lymphocyte counts were acutely reduced 3 hours post stimulation onset ($p = 0.006$ and $p = 0.011$, respectively; Fig. 3). Respective effect sizes were -0.60 for T cells and -0.54 for B cells, which are considered large effect sizes.”

Methods, page 15: “Differences between conditions were analyzed with Student’s t-tests for sleep and EEG data and with Wilcoxon-signed-rank tests for endocrine and immune parameters as these data were not normally distributed. Tests were one or two-sided, respectively, for directed vs. undirected hypothesis testing. To reduce Type 1 error with multiple comparisons of time series we clustered hormonal data into one-hour bins for statistical testing. A p-value < 0.05 was considered statistically significant. We calculated the effect size estimate r for the impact of SO stimulation on endocrine and immune parameters and followed Cohen’s criteria for interpretation of the sizes ($r = 0.1$, small; $r = 0.3$, medium; $r = 0.5$, large)⁵⁷. Correlations between the increase in SO activity during the stimulation interval and changes in endocrine and immunological parameters at the time points of significant effects were calculated with Spearman’s rho. The sample size was chosen based on previous studies that manipulated SWS non-specifically using pharmacological agents^{22,41}.”

Figure 2. Impact of auditory SO stimulation on cortisol, prolactin, and aldosterone levels. Means (\pm s.e.m.) of cortisol (a,b), prolactin (c) and aldosterone (d) levels calculated for one-hour bins. Grey area represents the 120-min stimulation period. * $p < 0.05$ for pairwise comparisons between the Stimulation condition (STIM, black) and the Sham condition (SHAM, white) with Wilcoxon tests, one sided. $n = 10-14$ (see Methods section for exact numbers).

Figure 3. Impact of auditory SO stimulation on lymphocyte counts. Means (\pm s.e.m.) of circulating T and B cell numbers (shown as difference from baseline). Grey area represents the 120-min stimulation period. * $p < 0.05$, ** $p < 0.01$ for pairwise comparisons between the Stimulation condition (STIM, black) and the Sham condition (SHAM, white) with Wilcoxon tests, one-sided. $n = 9$.

2. Blinding:

Were subjects blind to the condition? Was it verified, e.g. by asking in the morning, whether they were aware of the acoustic stimulation. This is important because the expectation of getting a “sleep support” may already affect the investigated endocrine markers. The randomization procedure should be mentioned.

Also related to the data analysis blinding is of importance. For example, were experimenters performing the sleep scoring blind to the condition?

Authors’ response:

Yes, subjects were blind to the condition. We asked the subjects in the morning whether they had heard the acoustic stimulation. Since only two out of 14 subjects reported having shortly noticed the stimulation a few times at the beginning of the night, it is unlikely that this has affected the results. Analysis of all data was performed blind to the condition. The randomization was performed in a semi-automated manner, ensuring that the order of conditions was balanced across subjects. We added this information to the manuscript as follows:

Methods, pages 11: “The two conditions were separated by at least two weeks (and not more than four weeks) and randomization was performed in a semi-automated manner, ensuring that the order of conditions was balanced across subjects. Subjects were blind to the condition, which was overall confirmed by the fact that only two out of 14 subjects reported having shortly noticed auditory stimuli at the beginning of the night. Analyses of EEG data, hormones and immune parameters were performed with the experimenter being unaware of the condition.

All participants had a regular sleep/wake rhythm, ...”

3. Specificity of the stimulation:

The authors write that their stimulation “distinctly increases slow oscillatory EEG activity”. However, the effects of the stimulation on other frequencies is not presented. Ideally a power spectrum for the duration of stimulation is presented for both conditions and compared statistically.

Moreover, slow wave activity (SWA) is strongly regulated. Thus, the question arises whether deepening sleep in the first period of sleep has an impact on subsequent sleep (e.g. second half of the night). Based on current models maybe decreased SWA during later periods might be expected. To address this point an analysis of the time course of SWA across the night would be needed. This is of particular importance since some of the effects are present after the stimulation.

Authors' response:

This is an important question. We have now calculated EEG power spectra for the stimulation interval. The auditory stimulation did only affect frequencies in the SO range. In addition, we have now also calculated the time course of SO activity across the entire night for one-hour bins. SO activity was only affected by the stimulation in the first two hours, i.e., during the stimulation interval, but not post stimulation. This speaks against the idea that the observed effects on hormones that occurred after the stimulation are due to a homeostatic counter-regulation of SO activity. We added this information to our manuscript as follows:

Results, page 5: "...as well as the amplitude of SOs ($p = 0.003$ and $p = 0.024$, respectively, for electrode position Fz; Fig. 1b,c). Other frequencies remained unchanged and SO activity was only affected during the stimulation interval but not afterwards (Supplementary Figs. 1, 2)."

Supplementary Figure 1: Auditory closed-loop stimulation selectively enhances slow oscillatory activity. Mean (\pm s.e.m.) spectral power during stimulation for electrode position Fz determined for NonREM epochs of the 120-min stimulation period for the Stimulation (black) and Sham condition (grey) for frequencies up to 30 Hz. The bottom panel indicates significance between the effects of the Stimulation and Sham condition (paired t-tests, two-sided). The insert shows the power in the SO peak frequency band (0.8-1.1 Hz); $**p < 0.01$. $n = 14$.

Supplementary Figure 2. Time course of slow oscillatory activity across the whole night. Mean (\pm s.e.m.) slow oscillatory (SO) activity across the whole night in one-hour intervals starting with stimulation onset. $*p < 0.05$ for pairwise comparisons between the Stimulation condition (STIM, black) and the Sham condition (SHAM, grey) with paired t-tests, two-sided. $n = 14$. Please notice that no SO activity could be calculated for the 8th hour as most subjects did not display any NonREM sleep at that time.

4. Direct relationship:

Individual differences in the stimulation effect on SWA as well as its effect on endocrine markers exists. It would be interesting to test whether a relationship between these factors exists (or not). Thus correlation analysis should be presented.

Authors' response:

We calculated correlations between the effect of the stimulation on SO activity during the stimulation interval and changes in endocrine and immunological parameters at time points of significant effects. We did not find any significant correlations. However, this null effect could be due to the low number of subjects, which does not provide high enough power for correlational analyses. We included this information now in the manuscript.

Results, pages 5f: "... are considered large effect sizes. We did not find any correlations between the increase in SO activity during the stimulation interval and changes in endocrine and immune parameters. However, for correlational analyses, the number of subjects was rather low and therefore this null effect might reflect a lack of statistical power."

Methods, page 15: "Correlations between the increase in SO activity during the stimulation interval and changes in endocrine and immunological parameters at the time points of significant effects were calculated with Spearman's rho. The sample size was chosen based on ..."

Minor points:

5. Expressions related to the quantification of slow waves might be confusing: Slow wave activity, SO activity, slow waves sleep is used. When is what adequate and what is the difference?

Authors' response:

We used the term slow wave activity (SWA) to refer to the frequency range between 0.5 and 4 Hz, whereas SO activity includes only frequencies ≤ 1 Hz. When referring to the results of the current study, we used the term SO activity since we modified only frequencies in the SO range. When citing other studies in which there was no differentiation between SO activity and SWA, we used the term SWA. SO activity and SWA both refer to the spectral power in the corresponding frequency range. In contrast, SWS refers to the deepest sleep stage, which is characterized by slow waves in the EEG, including the SOs. We now explained the terms more clearly and used this terminology in a stricter manner:

Abstract: "Here, we enhance SOs in men by closed-loop stimulation, i.e., by delivering tones in synchrony with endogenous SOs."

Introduction, page 3: "SWS is hallmarked by slow waves in the electroencephalogram (EEG), which have a frequency of 0.5-4 Hz and include the slow oscillation (SO) frequencies ≤ 1 Hz. Slow wave activity (i.e., the spectral power in the frequency range of 0.5-4 Hz) is associated with the co-ordinate release of immune-active hormones, specifically with a suppression of cortisol and an increase in prolactin, growth hormone (GH) and aldosterone levels, which provides an optimal endocrine milieu for supporting adaptive immune functions^{3,5-7}. ... However, so far, evidence for the relationship between slow wave or SO activity and the endocrine and immune functions of interest is solely based on correlational findings and, to the best of our knowledge, there is no direct proof of a causal role of SOs and SWS in this relationship. ... However, there have been no attempts to specifically strengthen SWS or SOs to boost its accompanying endocrine and immunological changes."

Discussion, page 9: "In such conditions, enhanced SO activity after closed-loop stimulation might well compensate for respective changes in hormonal release during sleep, ..."

6. The authors write “stimulation started with the onset of consolidated NonREM sleep”. How was this defined? Was this variable across subjects? Please provide a stimulation onset parameter. Moreover, stimulation was “terminated after 120 min”. Was this independent of the sleep structure of a subject, i.e. including waking and REM sleep? As a consequence the number of acoustic stimulation may vary a lot across subject. Please provide this data.

Authors’ response:

The stimulation was started 10 min after visual detection of 5 slow waves with an amplitude of at least $-80 \mu\text{V}$ within 30 s. After the first stimulation was applied, it was halted to observe the reaction of the subject. If no arousal was observed in the EEG, the stimulation was continued. This time point was defined as the onset of consolidated NonREM sleep. On average, the stimulation started 49 min after lights off (± 6.86 s.e.m.). The stimulation was always terminated 120 min after stimulation onset, independently of the sleep stage. Indeed, there was considerable variation in the number of stimulations among subjects. On average, 172 stimulations (including two acoustic stimuli) were delivered. The range was 52 – 307 stimulations. We had decided to terminate the stimulation in all subjects after 120 min, independently of the number of delivered stimulations, as otherwise we would have had different numbers of blood samplings among subjects, which would have complicated the statistical comparison. The high variability among subjects suggests that the effect of the stimulation might have been even more pronounced with more homogenous stimulation across subjects. We now added this additional data to our manuscript.

Results, page 4: “On average, the stimulation started 49 min (± 6.86 s.e.m.) after lights were turned off and on average 172 (± 24.3 s.e.m.) acoustic stimulations (each including two tones) were delivered in the 120-min interval. Confirming our previous studies¹² ...”

Methods, pages 10f: “... whenever an endogenous SO was identified electroencephalographically during NonREM sleep. Stimulation started with the onset of consolidated NonREM sleep (i.e., 10 minutes after visual detection of 5 slow waves with an amplitude $> 80 \mu\text{V}$ within 30 s in the EEG), and terminated after 120 min independently of the current sleep stage.”

7. Subject numbers vary for the different parameters. Please explain carefully.

Authors’ response:

For the first three subjects, we collected blood every 2.5 min to estimate which sampling rate would be optimal for observing changes in hormonal secretion. After an interim inspection of results, we decided that blood sampling every 5 min is enough to detect even rapid changes in hormonal levels. Since the amount of blood we can collect per subject is limited due to ethical considerations, reducing the sampling rate to 5 min during the stimulation period allowed us to draw blood also post stimulation until the end of the night in 15 min intervals. As we could not collect blood for the post-stimulation interval for the first 3 subjects (with a high blood sampling rate of 2.5 min during stimulation), we only have 11 subjects for blood samplings after the stimulation interval. For aldosterone, one subject had to be removed from the analysis because of unusually high aldosterone levels in both conditions, leading to $n = 13$ and $n = 10$ for aldosterone levels analyzed for the stimulation interval and the post-stimulation interval, respectively. Immune parameters were added after the first 5 subjects to see if there is any effect of the stimulation on hormones, which might induce changes in the analyzed immune parameters. Therefore, for immune parameters the sample

size is 9. Although this is a small sample size, the results for immune parameters are particularly robust, and we want to emphasize that they show the largest effect sizes of all the parameters in this study.

Methods, pages 15f: “The sample size was chosen based on previous studies that manipulated SWS non-specifically using pharmacological agents^{22,41}. For the first three subjects, we started with a blood sampling rate of 2.5 min for the stimulation interval, which precluded collecting more blood post stimulation due to ethical limitations in the amount of blood that can be collected per subject. For the other 11 subjects, we reduced the blood sampling rate during the stimulation interval to 5 min, as this was still high enough to detect even rapid changes in endocrine levels and allowed us to collect blood also post stimulation. One subject had to be excluded post-hoc from the analysis of aldosterone concentrations, as he showed unusually high levels of this hormone in both conditions. Immune parameters were collected in a subset of nine subjects.”

8. In the discussion about prolactin an endogenous rhythm in the slow wave frequency range is mentioned. Could that mean that the effects were not driven by changes in SO activity but rather direct effects occurred? If no correlation between SO activity and endocrine markers exists that might be an interpretation which should be discussed.

Authors’ response:

This is a very interesting idea and indeed, we cannot definitely answer from our human data whether the changes we observed are driven by the increase in SO activity or might reflect direct effects of the auditory signal at the hypothalamic level, especially since we did not find any correlation between stimulation-induced changes in SO activity and changes in hormone levels. Animal studies are needed here to measure the effect of auditory stimulation at hypothalamic level and explore whether there is a correlation between putative changes in hypothalamic rhythm and hormone levels. We now discuss this possible interpretation:

Discussion, page 7: “Interestingly, this activity exhibits an endogenous rhythm of 0.05-4 Hz²⁹ and, thus, might be particularly sensitive to exogenous stimulation of oscillations in the same frequency range. Animal studies would be helpful in this context to explore whether the observed changes are due to effects of the auditory stimulation on cortical SOs or rather reflect possible direct hypothalamic effects of the stimulation. Overall, the exact mechanisms of the stimulation-induced changes in endocrine activity are clearly in need of further investigation.”

9. Why was scoring not done according to the novel criteria of the AASM Manual? Including frontal channels in the scoring (as proposed in the manual) may reveal more SWS (under stimulation).

Authors’ response:

We decided to score based on the criteria of Rechtschaffen & Kales as they discriminate between sleep stages 3 and 4, allowing a more fine-grained analysis of SWS. We could have also scored according to the AASM manual and additionally differentiate between S3 and S4. However, there exists no normative data for this and this approach would also preclude a direct comparison to our previous study (Ngo et al., 2013. *Neuron*), which used the identical stimulation protocol. We performed some exploratory analyses in this regard and found AASM based scoring led to the expected increase of about 10 min for SWS (Moser et al., 2009. *Sleep*) but otherwise did not

essentially change the results of our power analysis. We have now changed the Method section to clarify this issue as follows:

Methods, page 12: "Sleep stages were determined off-line using EEG recordings from C3, C4, EOG, and EMG for subsequent 30 s epochs according to standard criteria⁵³. Scoring according to these criteria of Rechtschaffen & Kales allows a fine-grained analysis of SWS by discriminating between sleep stages S3 and S4, for which normative data exists⁵⁴."

10. Why was the "stimulation discontinued for 2.5 s"?

Authors' response:

The stimulation was discontinued for 2.5 s to induce SO trains that resemble spontaneously occurring SO. Endogenous SOs typically occur in trains of two to three SO cycles and to mimic this, we applied two tones followed by a "refractory" phase of 2.5 s.

Methods, page 13: "... the second stimulus then followed after a fixed interval of 1,075 ms. Then, stimulation was discontinued for 2.5 s, in order to mimic the typical temporal organization of spontaneous SOs occurring in trains of 2 or 3 succeeding waves⁵⁵, before the detection algorithm started again."

Reviewer #3 (Remarks to the Author):

This paper addresses a very important and novel topic likely to be of interest to a wide audience. The work reported directly addresses the question of whether a non-invasive non-pharmacological method that can increase the amplitude of cortical slow oscillations also stimulates extra-cortical brain mechanisms engaged by physiological SWS and the peripheral benefits of physiological SWS. To address this issue, the authors have completed a randomized controlled cross-over trial (effective acoustic stimulation versus a sham condition) in 14 healthy young men and obtained blood samples at frequent intervals throughout the night. GH, prolactin, cortisol and aldosterone were assayed on each sample. These hormones exert modulatory effects on immune function and therefore T and B cell counts were also measured.

MAJOR COMMENTS

1. The lack of stimulation of GH is a major negative finding that needs to be mentioned in the abstract and discussed further. The stimulation of GH release by SWS, particularly in healthy young men who were the participants in the present study, is the best-documented mechanistic connection between SWS and peripheral function. Further, there is abundant experimental evidence for a dose-response relationship between the amount of SWA and the increase in plasma GH levels. In contrast, the release of other hormones, including prolactin and aldosterone, and the inhibition of ACTH and cortisol, are temporally coincident with episodes of SWS, but a dose-response relationship has not been observed.

Authors' response:

The lack of an effect on GH was indeed surprising given that its association with SWA is very well established. Our primary idea was that this null effect reflects a ceiling effect as we examined young healthy men with already high GH levels. Another explanation could be that the higher frequencies within the SWA range are responsible for controlling GH release. We specifically enhanced the SOs frequency (≤ 1 Hz) in this study, mainly because a previous study from our lab had shown that SOs are associated with an improved response to vaccination (Lange et al., 2011. *J Immunol*). SWA includes also the delta wave range (1 – 4 Hz), which was not changed in the current study, but could be important in the control of GH. On the other hand, there are studies suggesting that GH secretion is not dependent on SWS as selective SWS deprivation did not affect GH levels and because dissociations between the occurrence of SWS and GH secretion have been found (Steiger et al., 1987. *Acta Endocrinol (Copenh)*; Born et al., 1988. *Psychoneuroendocrinology*; Davidson et al., 1991. *J Psychiatr Neurosci*). There could be a neuronal mechanism that couples the onset of sleep, of SWS and GH release, leading to the usually observed association between SWS and GH levels (Born et al., 1988. *Psychoneuroendocrinology*; Davidson et al. 1991. *J Psychiatr Neurosci*). This would explain why selectively enhancing SOs did not affect GH levels in our study. We now mention the lack of effect on GH in the Abstract and discuss it more deeply in the manuscript.

Abstract: "SWS is associated with a unique endocrine milieu comprising minimum cortisol and high prolactin, growth hormone (GH) and aldosterone levels, thereby presumably fostering efficient adaptive immune responses. ... T and B cell counts also decrease, likely reflecting the redistribution of these cells towards lymphoid tissues. GH remains unchanged."

Discussion, page 6f: “The lack of effect of the SO stimulation on GH was surprising, given that its release shows a clear association with EEG slow waves and is regulated most strongly by sleep^{6,19,20}. This lack might reflect a ceiling effect as the release of this peptide hormone is already at a diurnal maximum during early nocturnal SWS in healthy humans. Thus, an enforcing effect of auditory stimulation of SOs may become apparent only in conditions of shallowed sleep and accompanying reductions in GH levels, such as during aging^{21,22}. Another explanation could be that the release of GH is not controlled specifically by slow waves in the SO frequency range (i.e. ≤ 1 Hz), but by higher frequencies in the delta range (1 – 4 Hz), which were not altered by our approach. On the other hand, there is also evidence that GH secretion is not directly depending on SWS but rather on sleep onset because selective SWS did not affect GH levels and also dissociations between the occurrence of SWS and GH secretion have been found²³⁻²⁵. Based on these findings, it has been suggested that there is a neuronal mechanism coupling the onset of sleep, SWS and GH secretion, leading to the typically observed association between SWS and GH levels, in the absence of a direct causal effect of this sleep stage on GH secretion^{23,25}. This concept would conclusively explain why we did not observe changes in GH levels following selective enhancement of SOs.”

2. An increase in prolactin release is not seen “concomitantly to deepening SWS” (contrary to the statement in the Abstract) but rather occurs during the hour following the end of stimulation. Can you exclude a rebound effect where, when acoustic stimuli stop, dopaminergic activity is better suppressed and prolactin increases ?

Authors’ response:

We corrected this statement in the Abstract as follows:

“Stimulation acutely reduces nadir cortisol concentrations and, with a delay, enhances prolactin and aldosterone levels. T and B cell counts...”

We cannot exclude that the observed effect in prolactin levels reflects a rebound effect. However, from a purely descriptive point of view, prolactin was already enhanced during the stimulation period (see Supplementary Fig. 3), speaking for a slowly evolving effect. In addition, SO activity was not changed in the post-stimulation interval, which also speaks against a rebound effect. We added this information in the manuscript as follows:

Results, page 5: “... as well as the amplitude of SOs ($p = 0.003$ and $p = 0.024$, respectively, for electrode position Fz; Fig. 1b,c). Other frequencies remained unchanged and SO activity was only affected during the stimulation interval but not afterwards (Supplementary Figs. 1, 2).”

Discussion, page 7: “... leading to a unique uncoupling of cortisol and aldosterone secretion during this sleep stage. The effect of SO stimulation on prolactin levels developed gradually (Supplementary Fig. 3b). The pituitary release of prolactin is mainly regulated in a negative, i.e., suppressive manner ...”

Supplementary Figure 2. Time course of slow oscillatory activity across the whole night. Mean (\pm s.e.m.) slow oscillatory (SO) activity across the whole night in one-hour intervals starting with stimulation onset. * $p < 0.05$ for pairwise comparisons between the Stimulation condition (STIM, black) and the Sham condition (SHAM, grey) with paired t-tests, two-sided. $n = 14$. Please notice that no SO activity could be calculated for the 8th hour as most subjects did not display any NonREM sleep at that time.

Supplementary Figure 3. Time course of cortisol, prolactin, aldosterone and growth hormone levels with and without auditory SO stimulation. Means (\pm s.e.m.) of cortisol (a), prolactin (b), aldosterone (c), and growth hormone (d) levels calculated for one-hour bins starting from stimulation onset. Grey area represents the 120-min stimulation period. * $p < 0.05$ for pairwise comparisons between the Stimulation condition (STIM, black) and the Sham condition (SHAM, white) with Wilcoxon tests, one-sided. $n = 10-14$ (see Methods section for exact numbers). Please note in (a) the different scaling of the y-axis for cortisol levels during and post stimulation, respectively, which was applied because of the strong circadian rhythm of this hormone.

3. An increase in aldosterone is not seen “concomitantly to deepening SWS” (contrary to the statement in the Abstract) but rather occurs 3-6 hours after the stimulation has ended.

Authors’ response:

We corrected this statement in the Abstract as follows:

“Stimulation acutely reduces nadir cortisol concentrations and, with a delay, enhances prolactin and aldosterone levels. T and B cell counts...”

4. A reduction of cortisol levels was indeed observed during the first hour of stimulation in the active versus sham condition. However, the SEM for the mean cortisol levels in the sham condition (Fig. 2a) is 3-4 times larger than for the active condition, which could reflect outliers. Have you examined a possible order effect of the two conditions ? When was the sampling catheter inserted relative to the beginning of sampling ?

Authors’ response:

The data has been previously inspected for outliers. Although purely speculative, the smaller SEMs in the Stimulation condition could be a direct consequence of the stimulation, i.e., a floor effect which might have reduced the variance between subjects by entraining the occurrence of SWS. The order of conditions was balanced, precluding a systematic effect. Please refer also to our answer to comment #5 by the first reviewer. The catheter was inserted 1.5 hours before the first blood sampling, in order to allow the subject to recover from any stress the insertion of the catheter might have caused. We now included in the manuscript the details about the order of conditions and the time of inserting the catheter.

Methods, pages 10f: “... participated in this randomized, within-subject cross-over study. Each of two experimental conditions started at 21:00 h with preparing polysomnography and insertion of a catheter for blood sampling (approximately 1.5 hours before the first blood sampling). The subject was then allowed to sleep from 23:00 to 7:00 h. ... The two conditions were separated by at least two weeks (and not more than four weeks) and randomization was performed in a semi-automated manner, ensuring that the order of conditions was balanced across subjects.”

5. The BMI of the subjects should be reported. Adiposity suppresses GH release and this could be an explanation for the lack of effect on GH.

Authors’ response:

A BMI > 25 kg/m² was an exclusion criterion and the mean BMI was 23 kg/m² ± 2.14 s.d. We now report this in the Methods section on page 10:

“Fourteen physically and mentally healthy men (mean age 24 years ± 2.16 s.d., mean BMI 23 kg/m² ± 2.14 s.d.) participated in this randomized, within-subject cross-over study.”

SUGGESTIONS

6. A figure showing the entire period of blood sampling, concomitantly with the profiles of SO activity, should be used to report mean profiles of GH, cortisol, prolactin and aldosterone levels as well B/T cell counts in each condition.

Authors' response:

We now show the entire period of blood sampling and also calculated the time course of SO activity for the whole night in one-hour intervals. We could not calculate SO activity for the last hour since most subjects did not show any NonREM sleep during this interval. We also have no baseline levels for SO activity since the stimulation started already with the onset of consolidated NonREM sleep. Therefore, we decided to show the results of the hormonal data and of the SO activity in separate graphs (see Supplementary Figs. 2 and 3 shown above).

7. In Figure 1, it is unclear whether tones were delivered when SO were present in NREM sleep (including N2). Panel 1c reports mean SO activity and amplitude for epochs of SWS (N3, N2 excluded). The mean number of SWS epochs used in the calculation of the data shown in 1c should be reported in the legend for each condition.

Authors' response:

The tones were delivered during NonREM sleep including N2. However, we presented the results for SWS epochs (N3) because most of the stimulations occurred during SWS. The average number of SWS epochs used for this calculation was 63 and 74, respectively, for the Stimulation and the Sham condition with no significant difference between conditions. Following the reviewer's suggestion, we now decided to show the results for the entire NonREM period as more epochs are included in these analyses (158 and 169, respectively). We adapted the manuscript accordingly:

Figure 1. Auditory stimulation phase-locked to endogenous SO peaks boosts SO activity. (a) Setup: Upon online detection of an endogenous SO in the frontal EEG signal during non-rapid eye movement (NonREM) sleep, two tones (50 ms, pink noise, 50 dB SPL) were delivered via in-ear headphones with an inter-stimulus interval of 1.075 s to coincide with two consecutive SO peaks. In the Sham condition, time points of stimulation were marked, but no stimuli were presented. See Methods section for further details. (b) Mean (\pm s.e.m.) EEG signal recorded from a frontal (Fz) electrode position during NonREM sleep (S2, S3 and S4) in the 120-min stimulation period, time-locked to the first of the two tones ($t = 0$) for the Stimulation (STIM, black) and Sham condition (SHAM, grey). (c) Mean (\pm s.e.m.) spectral power in the SO peak frequency band (0.8-1.1 Hz) and SO

amplitude recorded from electrode position Fz and determined for NonREM sleep epochs of the 120-min stimulation period. The average number of NonREM sleep epochs used for this calculation was 158 and 169, respectively, for the Stimulation and the Sham condition. (There was no significant difference in the number of epochs between conditions, $p = 0.123$). * $p < 0.05$, ** $p < 0.01$ for pairwise comparisons between the Stimulation condition (STIM, black) and the Sham condition (SHAM, grey) with paired t-tests, one-sided. $n = 14$.

8. *The statistical analysis needs to be repeated using two-tail tests since the result opposite to the hypothesis (for example: acoustic stimulation stimulates sympathetic nervous activity, preventing concomitant GH & prolactin release ??) cannot be excluded.*

Authors' response:

We used one-tailed tests for analyzing the effects of the stimulation on endocrine and immune parameters because we had clear *a priori* and literature-based hypotheses about the direction of the effects and the statistical hypotheses were thus unidirectional. Our interest in applying the closed-loop stimulation technique was to induce a specific hormonal milieu that is present during SWS. Therefore, changes opposite to the predicted direction were not only very unlikely (we knew, for example, from our previous study that the stimulation does not induce arousals and was thus unlikely to activate the stress axes) but also not in the center of our interest. We are, thus, still convinced that applying one-tailed tests is appropriate (see also Ludbrook, 2013. *Clinical and Experimental Pharmacology and Physiology*), and prefer to follow a suggestion by reviewer 2 regarding this issue, to also indicate in the Figure legends whether a two or one-sided test was applied. We also show exact p-values in the manuscript and with this information the reader can easily infer the corresponding p-value for a two-tailed test if desired.

9. *Differences in temporal profiles between conditions (active versus sham) should be analysis by ANOVA for repeated measures with condition as factor, time as repeated measure and the interaction time x condition.*

Authors' response:

We thank the reviewer for raising this important point. Please note for several time points, the hormonal and immune data were not normally distributed (even after logarithmic or square root transformation) which precludes straightforward application of ANOVA. We therefore have now employed non-parametric statistical analyses (Wilcoxon tests). To reduce the inflation of the Type I error, we clustered the data and analyzed them only for one-hour bins, which comes closer to running an ANOVA including the Factors "Condition" and "Time" than our previous approach (please, also refer to our answer to comment #1 by reviewer 2). Using this statistical approach, the results remain essentially the same as those previously reported. We adapted the manuscript accordingly:

Results, page 5: "Stimulation distinctly reduced blood cortisol concentrations during the first hour after stimulation onset ($p = 0.024$, Fig. 2a,b, Supplementary Fig. 3a), although levels were already rather low during this early-night interval. The reduction was visible already 5 min post stimulation onset and averaged 15% in the first hour of stimulation (Fig. 2a). Prolactin levels were increased in the Stimulation compared to the Sham condition during the 3rd hour after stimulation onset ($p = 0.038$; Fig. 2c, Supplementary Fig. 3b). Levels of aldosterone were increased during the 4th and 8th hour after stimulation onset ($p = 0.014$ and 0.037 , respectively; Fig. 2d, Supplementary Fig. 3c). GH remained unchanged (Supplementary Fig. 3d). Effect sizes (r) were -0.37 for cortisol, 0.38 for prolactin, and 0.49 and 0.40 respectively, for aldosterone, which reflect medium-sized effects. Blood

T and B lymphocyte counts were acutely reduced 3 hours post stimulation onset ($p = 0.006$ and $p = 0.011$, respectively; Fig. 3). Respective effect sizes were -0.60 for T cells and -0.54 for B cells, which are considered large effect sizes.”

Methods, page 15: “Differences between conditions were analyzed with Student’s t-tests for sleep and EEG data and with Wilcoxon-signed-rank tests for endocrine and immune parameters as these data were not normally distributed. Tests were one or two-sided, respectively, for directed vs. undirected hypothesis testing. To reduce Type 1 error with multiple comparisons of time series we clustered hormonal data into one-hour bins for statistical testing. A p-value < 0.05 was considered statistically significant. We calculated the effect size estimate r for the impact of SO stimulation on endocrine and immune parameters and followed Cohen’s criteria for interpretation of the sizes ($r = 0.1$, small; $r = 0.3$, medium; $r = 0.5$, large)⁵⁷. Correlations between the increase in SO activity during the stimulation interval and changes in endocrine and immunological parameters at the time points of significant effects were calculated with Spearman’s rho. The sample size was chosen based on previous studies that manipulated SWS non-specifically using pharmacological agents^{22,41}.”

Figure 2. Impact of auditory SO stimulation on cortisol, prolactin, and aldosterone levels. Means (\pm s.e.m.) of cortisol (a,b), prolactin (c) and aldosterone (d) levels calculated for one-hour bins. Grey area represents the 120-min stimulation period. * $p < 0.05$ for pairwise comparisons between the Stimulation condition (STIM, black) and the Sham condition (SHAM, white) with Wilcoxon tests, one sided. $n = 10-14$ (see Methods section for exact numbers).

Figure 3. Impact of auditory SO stimulation on lymphocyte counts. Means (\pm s.e.m.) of circulating T and B cell numbers (shown as difference from baseline). Grey area represents the 120-min stimulation period. * $p < 0.05$, ** $p < 0.01$ for pairwise comparisons between the Stimulation condition (STIM, black) and the Sham condition (SHAM, white) with Wilcoxon tests, one-sided. $n = 9$.

Reviewers' comments:

Reviewer #1 (Remarks to the Author):

The authors have been somewhat responsive to my comments but I remain concerned about the lack of an adequate control intervention. In essence there is no control for auditory stimulation. It is important for the field that an adequate control intervention is developed and implemented in these experiments. The arguments put forward by the Authors are in my view weak. Auditory stimulation which does not increase SOs but either leaves them unaffected or reduces SOs should have no effect on the endocrine/immune responses or an effect opposite to the effects of enhancing SOs. The authors have not even implemented a statistical control (i.e. controlling for the number of acoustic stimuli). In addition, the authors now report that there are no significant correlation between SO's and the endocrine responses and thereby provide evidence against a causal role of SOs. It'd be interesting to compute the correlation between the number auditory stimuli (which varied widely across individuals) and the endocrine responses.

Although the statistics have improved somewhat, non-parametric ANOVA was not applied. One sided testing is also not well justified.

I am now also curious about the newly reported 'effect size r'. In general this is an effect size derived from correlations and this will have to be explained.

Reviewer #2 (Remarks to the Author):

The authors carefully addressed the reviewers concerns. The revised manuscript (and supplementary materials) now includes additional information allowing the reader to judge the specificity and significance of the observed effects. With the exception of the following minor remaining remarks, the reviewer is satisfied with the revisions.

1-sided vs. 2-sided: In the reviewers opinion this is not consistently applied. For example, based on their previous studies, the authors must have expected an increase in SO power after stimulation. Nevertheless, here two-sided tests were used. This gives the impression that where 2-sided tests revealed a significant result 2-sided tests were applied... 1-sided testing should be mentioned in the discussion as a limitation, i.e. some of their findings only reached significance when using 1-sided tests.

Supplementary Figure 1: y-axis label is unclear. Looks like on the logarithmic scale microVolts^{2} is presented (and not percentage). The inset on the other hand may be represented in % - though it is unclear % of what. This should be clearly described in the Figure legend.

Supplementary Figure 2: Similar to Figure S1 it is unclear what "normalized" means. In addition, is the unit here really Hz^{-1} ? Not %.

Correlations: Studies with a similar number of subjects (also from their own group, e.g. Ngo et al., Neuron 2013) have found correlations – and it was not stated that these correlations

might have been "randomly observed". Thus, the statement "However, for correlational analyses,..." should be omitted. Given that there was quite some variation in the number of stimuli that were applied (as now included in the manuscript), and, as a result in the effect on SO power, it is surprising that this does not show up in respective hormonal changes. The reviewer would rather prefer that this lack of a relationship is discussed.

Reviewer #3 (Remarks to the Author):

The authors have carefully revised the manuscript and addressed most of the issues the issues raised by the reviewers. Areas of uncertainty regarding the main claims remain but they are explicitly addressed in the discussion. The statistical analysis has been considerably strengthened. The use of one-tail testing is a weakness but it is clearly acknowledged.

The stimulation of SO power and amplitude illustrated in Figure 1 is very modest, under 10% on average. This may be in the range of day to day variability in a given subject. This should also be discussed.

In detail, the following changes have been introduced regarding the referees' comments (highlighted in yellow):

Reviewer #1 (Remarks to the Author):

1. The authors have been somewhat responsive to my comments but I remain concerned about the lack of an adequate control intervention. In essence there is no control for auditory stimulation. It is important for the field that an adequate control intervention is developed and implemented in these experiments. The arguments put forward by the Authors are in my view weak. Auditory stimulation which does not increase SOs but either leaves them unaffected or reduces SOs should have no effect on the endocrine/immune responses or an effect opposite to the effects of enhancing SOs. The authors have not even implemented a statistical control (i.e. controlling for the number of acoustic stimuli). In addition, the authors now report that there are no significant correlation between SO's and the endocrine responses and thereby provide evidence against a causal role of SOs. It'd be interesting to compute the correlation between the number auditory stimuli (which varied widely across individuals) and the endocrine responses.

Authors' response:

We want to thank the reviewer for suggesting to implement a statistical control, which we have now implemented as presented below. Nevertheless, we would first like to point out that, in our experience, adding another 'acoustic stimulation' control as described by the reviewer would not significantly add to our data and we are confident that a Sham control group is the most adequate one. Previous experiments have shown that control groups that, as suggested by the reviewer, were randomly presented with stimuli comparable to the stimuli used in our study (i.e. same characteristics and volume) also showed significant increases in SO power to some extent (see Weigenand et al., 2016. *Eur J Neurosci*). Furthermore, this random application of stimuli during NREM sleep produced clear side effects: it changed other sleep parameters (unrelated to SWS), it was noticed during the night by half of the subjects and even perceived as unpleasant by some of the participants. In our previous study (Ngo et al., 2013. *Neuron*), we included a control group where stimuli were applied out-of-phase with endogenous SOs. This condition suppressed the development of SO trains, however SO power was not robustly reduced and there was a pronounced increase of power in the delta range (i.e. 1 – 4 Hz). Hence, it is absolutely not a trivial task to develop a control in which the same auditory stimuli do not affect or even robustly reduce SOs. Also, developing and executing such control *per se* would be a highly effort and time-consuming endeavour (conducting the original study took more than two years) which we feel is beyond the scope of the present revision. We agree with the reviewer that it is important to go into this direction and to ultimately develop additional experimental controls, but this is work for several additional studies. We have therefore opted here for the reviewer's suggestion of a statistical control.

To substantiate our notion that the observed changes in endocrine/immune parameters were not induced by the stimuli *per se*, but rather by the SOs, we performed additional correlational analyses. Specifically, we have now performed hierarchical linear regression analyses including the first four one-hour bins of hormonal and SO measurements. Endocrine/immunological variables were included as dependent variable, SO power as independent factor, and these analyses included the factor "Number of applied auditory stimuli" to control for potential contributions of the auditory stimulation *per se*. The analyses were extended to the four-hour interval because this is the time of predominant SWS, and we assumed that variance of SO activity averaged across the stimulation

interval (as used in our previous correlational analyses) is too low for detecting significant associations (see Ngo et al. 2013, *Neuron*, for a comparable lack of correlation when averaged SO activity is used for calculations). The analyses were corrected for the factor “Time bin”, i.e. the four one-hour bins, to exclude variability explained by including four bins per subject (see e.g. Bland, M. *An Introduction to Medical Statistics*. 3rd edn, (Oxford University Press, 2000)). A distribution independent bootstrapping procedure with 10,000 samples was used for the regressions because data were not normally distributed. These analyses revealed a negative correlation between cortisol levels and SO power, which remained significant after controlling for the number of applied stimuli ($\beta = -0.397$, $p = 0.019$). SO activity also predicted aldosterone levels with a time lag of two hours, which reflects the delayed impact of the stimulation on this parameter ($\beta = 0.577$, $p = 0.045$, controlled for number of applied stimuli). These findings point to a direct causal role of SOs in inducing the observed changes, and speak against the possibility that the auditory stimuli *per se* affected cortisol and aldosterone levels independently of the increase in SO activity. We did not find correlations between SO activity and lymphocytes, possibly because effects on these parameters were less directly mediated by SOs, including mediation by other peripheral systems.

We have also run hierarchical regression analyses with the number of applied auditory stimuli as a predictor variable for endocrine/immune parameters as suggested by the reviewer. There was no association between these parameters, again speaking against the possibility that the auditory stimuli *per se* induced the changes. In contrast, the number of auditory stimuli was significantly associated with SO activity when employing the regression analysis ($\beta = 0.420$, $p = 0.004$). These new analyses are now included and discussed in the manuscript:

Results, page 6: “We performed explorative regression analyses to further examine associations between SO activity and concentrations of the endocrine and immune parameters of interest, and also to assess possible confounding effects of the auditory stimuli *per se* (see Methods section for details). SO activity was a significant predictor of cortisol levels ($\beta = -0.397$, $p = 0.019$) and, with a time lag of two hours, of aldosterone levels, with the time lag reflecting the delayed impact of stimulation on this parameter ($\beta = 0.577$, $p = 0.045$). We did not find correlations between SO activity and lymphocytes, possibly because effects on these parameters were less direct. The number of applied auditory stimuli significantly predicted SO activity ($\beta = 0.420$, $p = 0.004$), but was not associated with endocrine/immune parameters, which rules out that the auditory stimuli *per se* substantially contributed to the hormonal and immunological effects of the SO stimulation.”

Discussion, page 10: “SO activity was a significant predictor of cortisol and aldosterone levels, despite controlling for the number of applied stimuli. Also, the number of auditory stimuli *per se*, although predicting SO activity, did not predict concentrations of the endocrine or immune parameters of interest. These findings point to a direct causal role of SOs in inducing the observed changes, and speak against the possibility that the auditory stimuli *per se* affected the endocrine/immune parameters independently of the increase in SO activity. Nevertheless, we cannot exclude that the auditory stimulation additionally affected other parameters...”

Methods, page 17: “Correlations of mean SO activity during the stimulation interval with endocrine/immune parameters during time intervals of significant effects and with the number of applied auditory stimuli were calculated using Spearman’s rho. The correlations remained non-significant ($r < 0.3$, $p > 0.289$), presumably due to the low between-subject variance in SO activity during the stimulation interval (see ¹² for a comparable lack of correlation) and were not reported here in detail. Hence, at a second step, we performed hierarchical linear regression analyses including the

parameters of interest over an extended period, i.e., the first four one-hour bins post stimulation onset, which is the time with predominant SWS. These analyses included SO activity as predictor variable and the different hormone/lymphocyte measures as dependent variables, while correcting for the factor “Time bin” (to control for variance explained by inclusion of the four time bins per subject). To control for possible contributions of the auditory stimuli *per se*, the analyses were additionally corrected for the factor “Number of applied auditory stimuli”. Further analyses were performed with the number of auditory stimuli as predictor variable for SO activity and for endocrine/immune parameters. A distribution independent bootstrapping procedure with 10,000 samples was used for the regressions because endocrine/immune parameters and the number of applied stimuli were not normally distributed. To account for delayed effects of the stimulation on aldosterone and lymphocytes, we performed the analyses including a delay (time lag) of the SO activity of 0, 1, 2, and 3 hours relative to these peripheral parameters.

The sample size was chosen based on previous studies...”

2. Although the statistics have improved somewhat, non-parametric ANOVA was not applied. One sided testing is also not well justified.

Authors’ response:

Non-parametric ANOVAs can be applied for one-way designs. However, two-way non-parametric ANOVAs cannot be conclusively interpreted because interactions are based on comparisons of differences between ranks, so there is not a non-parametric counterpart for factorial repeated measures designs (see Field, A. *Discovering statistics using SPSS*. 4th edn. (Sage Publications, 2013)). Other types of distribution independent tests, like bootstrapping methods, are to our knowledge not established for ANOVAs with repeated measures including more than one factor. Therefore, applying Wilcoxon tests while correcting for repeated measurements by clustering data is, in our view, the most appropriate and well-established statistical approach to the current data.

However, to further reduce the risk of Type 1 error, we have now decided to report two-sided tests for all analyses. Results for cortisol, aldosterone, T cells and B cells do not change by employing two-sided tests, and the effect on prolactin is the only one that now fails to reach statistical significance when clustering the data into one-hour bins. We have now adapted the manuscript accordingly:

Abstract: “Stimulation intensifies the hormonal milieu characterizing SWS (mainly by further reducing cortisol and increasing aldosterone levels) and reduces T and B cell counts, likely reflecting a redistribution of these cells to lymphoid tissues. GH remains unchanged.”

Introduction, page 4: “We show that selectively enhancing SOs through auditory stimulation intensifies the immune-supportive hormonal milieu present during SWS (mainly by further reducing cortisol and increasing aldosterone levels) and supports the extravasation of T and B lymphocytes.”

Results, page 5: “Aldosterone levels were significantly increased in the Stimulation compared to the Sham condition during the 4th hour after stimulation onset ($p = 0.028$; Fig. 2c, Supplementary Fig. 3b). An increase in aldosterone levels during the 8th hour and in prolactin levels during the 3rd hour after stimulation onset approached significance ($p \leq 0.075$; Fig. 2c,d, Supplementary Fig. 3b,c), with exploratory analyses revealing enhanced prolactin levels after stimulation at two of the 15-min samplings (150 min and 165 min post stimulation onset, respectively, $p \leq 0.050$). GH remained unchanged (Supplementary Fig. 3d).”

Discussion, page 6ff: “We show here that deepening sleep by EEG closed-loop auditory stimulation of SOs robustly decreases levels of the anti-inflammatory hormone cortisol, increases levels of aldosterone and tends to increase also prolactin levels, thus enforcing the immune-supportive hormonal milieu that is unique to SWS³.

... Considering that our method is highly specific in selectively enhancing SOs without affecting other sleep parameters, our findings suggest a causal role of SOs in supporting the overall pro-inflammatory hormonal milieu characterizing SWS.

... The effect of SO stimulation on prolactin levels developed gradually (Supplementary Fig. 3c) and failed to reach significance after clustering data of the frequent blood samplings into one-hour bins. The pituitary release of prolactin is mainly regulated in a negative, i.e., suppressive manner by dopaminergic activity of the arcuate nucleus of the hypothalamus. Interestingly, this activity exhibits an endogenous rhythm of 0.05-4 Hz²⁹ and, thus, might be sensitive to exogenous stimulation of oscillations in the same frequency range. This idea is supported by our findings. However, the effect of the stimulation on prolactin was not robust, and also in light of conflicting findings as to an association of prolactin release with specific sleep stages²⁰, the role of SOs in the regulation of this hormone needs to be further scrutinized.”

Methods, page 16: “Differences between conditions were analyzed with two-sided Student’s t-tests for sleep and EEG data and with two-sided Wilcoxon-signed-rank tests for endocrine and immune parameters as these data were not normally distributed.”

3. I am now also curious about the newly reported 'effect size r'. In general this is an effect size derived from correlations and this will have to be explained.

Authors’ response:

The reported effect size r is the equivalent to Cohen’s d for non-parametric tests (see Field, A. *Discovering statistics using SPSS*. 4th edn. (Sage Publications, 2013)). It is calculated as follows: Z score (test statistic of Wilcoxon tests) divided by the square root of the number of subjects over both experimental conditions. We added this information to the manuscript as follows:

Methods, page 16: “For the impact of SO stimulation on endocrine and immune parameters, we calculated the effect size estimate r , which is used instead of Cohen’s d for non-parametric tests, with the following criteria for interpretation of the sizes: $r = 0.1$, small; $r = 0.3$, medium; $r = 0.5$, large⁵⁷.”

Reviewer #2 (Remarks to the Author):

The authors carefully addressed the reviewers concerns. The revised manuscript (and supplementary materials) now includes additional information allowing the reader to judge the specificity and significance of the observed effects. With the exception of the following minor remaining remarks, the reviewer is satisfied with the revisions.

1. 1-sided vs. 2-sided: In the reviewers opinion this is not consistently applied. For example, based on their previous studies, the authors must have expected an increase in SO power after stimulation. Nevertheless, here two-sided tests were used. This gives the impression that where 2-sided tests revealed a significant result 2-sided tests were applied... 1-sided testing should be mentioned in the discussion as a limitation, i.e. some of their findings only reached significance when using 1-sided tests.

Authors' response:

There might have been a misunderstanding as SO power in the stimulation interval was analyzed with one-sided tests (see Figure 1 of the previous manuscript version). However, this is not relevant anymore because, considering a related comment (#2) by Reviewer #1, we have now decided to report two-sided tests for **all** analyses in order to avoid the limitations of one-sided testing. Results for cortisol, aldosterone, T cells and B cells do not change by employing two-sided tests, and the effect on prolactin is the only one that now fails to reach statistical significance when clustering the data into one-hour bins. We have now adapted the manuscript accordingly:

Abstract: "Stimulation intensifies the hormonal milieu characterizing SWS (mainly by further reducing cortisol and increasing aldosterone levels) and reduces T and B cell counts, likely reflecting a redistribution of these cells to lymphoid tissues. GH remains unchanged."

Introduction, page 4: "We show that selectively enhancing SOs through auditory stimulation intensifies the immune-supportive hormonal milieu present during SWS (mainly by further reducing cortisol and increasing aldosterone levels) and supports the extravasation of T and B lymphocytes."

Results, page 5: "Aldosterone levels were significantly increased in the Stimulation compared to the Sham condition during the 4th hour after stimulation onset ($p = 0.028$; Fig. 2c, Supplementary Fig. 3b). An increase in aldosterone levels during the 8th hour and in prolactin levels during the 3rd hour after stimulation onset approached significance ($p \leq 0.075$; Fig. 2c,d, Supplementary Fig. 3b,c), with exploratory analyses revealing enhanced prolactin levels after stimulation at two of the 15-min samplings (150 min and 165 min post stimulation onset, respectively, $p \leq 0.050$). GH remained unchanged (Supplementary Fig. 3d)."

Discussion, page 6ff: "We show here that deepening sleep by EEG closed-loop auditory stimulation of SOs robustly decreases levels of the anti-inflammatory hormone cortisol, increases levels of aldosterone and tends to increase also prolactin levels, thus enforcing the immune-supportive hormonal milieu that is unique to SWS³.

... Considering that our method is highly specific in selectively enhancing SOs without affecting other sleep parameters, our findings suggest a causal role of SOs in supporting the overall pro-inflammatory hormonal milieu characterizing SWS.

... The effect of SO stimulation on prolactin levels developed gradually (Supplementary Fig. 3c) and failed to reach significance after clustering data of the frequent blood samplings into one-hour bins.

The pituitary release of prolactin is mainly regulated in a negative, i.e., suppressive manner by dopaminergic activity of the arcuate nucleus of the hypothalamus. Interestingly, this activity exhibits an endogenous rhythm of 0.05-4 Hz²⁹ and, thus, might be sensitive to exogenous stimulation of oscillations in the same frequency range. This idea is supported by our findings. However, the effect of the stimulation on prolactin was not robust, and also in light of conflicting findings as to an association of prolactin release with specific sleep stages²⁰, the role of SOs in the regulation of this hormone needs to be further scrutinized.”

Methods, page 16: “Differences between conditions were analyzed with two-sided Student’s t-tests for sleep and EEG data and with two-sided Wilcoxon-signed-rank tests for endocrine and immune parameters as these data were not normally distributed.”

2. *Supplementary Figure 1: y-axis label is unclear. Looks like on the logarithmic scale microVolts² is presented (and not percentage). The inset on the other hand may be represented in % - though it is unclear % of what. This should be clearly described in the Figure legend.*

Authors’ response:

We apologize for this lack of clarity in the y-axis label. The power analyses used normalized values to control for inter-individual variability. For normalization, the power spectrum for each subject was divided by its cumulative power up to 30 Hz (i.e. the sum of the spectral values multiplied by the frequency resolution). The unit of the cumulative power, i.e. the area under the curve, corresponds to $\mu\text{V}^2 \times \text{Hz}$, hence the normalization yields the illustrated unit of Hz^{-1} . Multiplying the normalized values by 100 yields percentage values which, for reasons of clarity, we have now omitted. The y-axis for the graph shown in Supplementary Figure 1 (including the insert) is the same as in Figure 1c and Supplementary Figure 2 and we now unified the label to be consistent across all Figures. We now also explain the normalization procedure in more detail in the Methods section and additionally mention it in the Figure legends:

Methods, page 15: “To account for individual variability, we normalized the power spectrum for each subject by dividing it by its cumulative power up to 30 Hz. The unit of the cumulative power, i.e. the area under the curve, corresponds to $\mu\text{V}^2 \times \text{Hz}$, hence the normalization yields the unit Hz^{-1} .”

Legend of Figure 1: “...(c) Mean (\pm s.e.m.) normalized spectral power in the SO peak frequency band (0.8-1.1 Hz) and SO amplitude recorded from electrode position Fz and determined for NonREM sleep epochs of the 120-min stimulation period. The average number of NonREM sleep epochs used for this calculation was 158 and 169, respectively, for the Stimulation and the Sham condition. (There was no significant difference in the number of epochs between conditions, $p = 0.123$). For normalization individual spectra were divided by the cumulative power (up to 30 Hz) $**p < 0.01$, $*p < 0.05$, for pairwise comparisons between the Stimulation condition (STIM, black) and the Sham condition (SHAM, grey) with paired t-tests, two-sided. $n = 14$.”

Legend of Supplementary Figure 1: “**Supplementary Figure 1: Auditory closed-loop stimulation selectively enhances slow oscillatory activity.** Mean (\pm s.e.m.) normalized spectral power during stimulation for electrode position Fz determined for NonREM epochs of the 120-min stimulation period for the Stimulation (black) and Sham condition (grey) for frequencies up to 30 Hz. For normalization individual spectra were divided by the cumulative power (up to 30 Hz). The bottom panel indicates significance between the effects of the Stimulation and Sham condition (paired t-

tests, two-sided). The insert shows the **normalized** power in the SO peak frequency band (0.8-1.1 Hz); ****p < 0.01. n = 14.**"

3. *Supplementary Figure 2: Similar to Figure S1 it is unclear what "normalized" means. In addition, is the unit here really power Hz⁻¹? Not %.*

Authors' response:

The y-axis label for Supplementary Figure 2 is the same as for Figure 1c and Supplementary Figure 1. It represents the normalized power, as described in our answer to the previous comment. We now adapted the legend of this figure accordingly to the other figures:

Legend of Supplementary Figure 2: **"Supplementary Figure 2. Time course of slow oscillatory activity across the whole night. Mean (\pm s.e.m.) **normalized** slow oscillatory (SO) activity across the whole night in one-hour intervals starting with stimulation onset. **For normalization individual spectra were divided by the cumulative power (up to 30 Hz).** *p < 0.05 for pairwise comparisons between the Stimulation condition (STIM, black) and the Sham condition (SHAM, grey) with paired t-tests, two-sided. n = 14. Please notice that no SO activity could be calculated for the 8th hour as most subjects did not display any NonREM sleep at that time."**

4. *Correlations: Studies with a similar number of subjects (also from their own group, e.g. Ngo et al., Neuron 2013) have found correlations – and it was not stated that these correlations might have been "randomly observed". Thus, the statement "However, for correlational analyses,..." should be omitted. Given that there was quite some variation in the number of stimuli that were applied (as now included in the manuscript), and, as a result in the effect on SO power, it is surprising that this does not show up in respective hormonal changes. The reviewer would rather prefer that this lack of a relationship is discussed.*

Authors' response:

Please note, the significant correlations reported in our previous study (Ngo et al., 2013. *Neuron*) were on parameters unrelated to SO power. In that study we did also not find any correlations between SO power (averaged across the stimulation interval) and the dependent variable (memory retention). The lack of correlation in the previous study, which is likewise found in the present study, is likely due to the low variance in SO power averaged across the stimulation interval, reflecting a ceiling effect in our young healthy subjects. We therefore have now performed hierarchical linear regression analyses including the first four one-hour bins of hormonal and SO measurements, which is the time interval when SWS is predominant. Endocrine/immunological variables were included as dependent variable, SO power as independent factor, and analyses were corrected for the factor "Time bin" (i.e. the four one-hour bins, to exclude variability that is explained by including four bins per subject, see e.g. Bland, M. An Introduction to Medical Statistics. 3rd edn, (Oxford University Press, 2000)). To control for possible contributions of the auditory stimuli *per se* (see comment #1 by Reviewer #1), the analyses were additionally corrected for the factor "Number of applied auditory stimuli". A distribution independent bootstrapping procedure with 10,000 samples was used for the regressions because data were not normally distributed. These analyses indeed revealed a negative correlation between cortisol levels and SO power, even with correction for the number of applied auditory stimuli ($\beta = -0.397$, $p = 0.019$). SO activity also predicted aldosterone levels with a time lag of two hours, which reflects the delayed impact of the stimulation on this parameter ($\beta = 0.577$, $p =$

0.045, controlled for number of applied stimuli). We did not find correlations between SO activity and lymphocytes, possibly because effects on these parameters were less directly mediated by SOs, including mediation by other peripheral systems.

These new analyses are now included and discussed in the manuscript:

Results, page 6: “We performed explorative regression analyses to further examine associations between SO activity and concentrations of the endocrine and immune parameters of interest, and also to assess possible confounding effects of the auditory stimuli *per se* (see Methods section for details). SO activity was a significant predictor of cortisol levels ($\beta = -0.397$, $p = 0.019$) and, with a time lag of two hours, of aldosterone levels, with the time lag reflecting the delayed impact of stimulation on this parameter ($\beta = 0.577$, $p = 0.045$). We did not find correlations between SO activity and lymphocytes, possibly because effects on these parameters were less direct. The number of applied auditory stimuli significantly predicted SO activity ($\beta = 0.420$, $p = 0.004$), but was not associated with endocrine/immune parameters, which rules out that the auditory stimuli *per se* substantially contributed to the hormonal and immunological effects of the SO stimulation.”

Discussion, page 10: “SO activity was a significant predictor of cortisol and aldosterone levels, despite controlling for the number of applied stimuli. Also, the number of auditory stimuli *per se*, although predicting SO activity, did not predict concentrations of the endocrine or immune parameters of interest. These findings point to a direct causal role of SOs in inducing the observed changes, and speak against the possibility that the auditory stimuli *per se* affected the endocrine/immune parameters independently of the increase in SO activity. Nevertheless, we cannot exclude that the auditory stimulation additionally affected other parameters...”

Methods, page 17: “Correlations of mean SO activity during the stimulation interval with endocrine/immune parameters during time intervals of significant effects and with the number of applied auditory stimuli were calculated using Spearman’s rho. The correlations remained non-significant ($r < 0.3$, $p > 0.289$), presumably due to the low between-subject variance in SO activity during the stimulation interval (see ¹² for a comparable lack of correlation) and were not reported here in detail. Hence, at a second step, we performed hierarchical linear regression analyses including the parameters of interest over an extended period, i.e., the first four one-hour bins post stimulation onset, which is the time with predominant SWS. These analyses included SO activity as predictor variable and the different hormone/lymphocyte measures as dependent variables, while correcting for the factor “Time bin” (to control for variance explained by inclusion of the four time bins per subject). To control for possible contributions of the auditory stimuli *per se*, the analyses were additionally corrected for the factor “Number of applied auditory stimuli”. Further analyses were performed with the number of auditory stimuli as predictor variable for SO activity and for endocrine/immune parameters. A distribution independent bootstrapping procedure with 10,000 samples was used for the regressions because endocrine/immune parameters and the number of applied stimuli were not normally distributed. To account for delayed effects of the stimulation on aldosterone and lymphocytes, we performed the analyses including a delay (time lag) of the SO activity of 0, 1, 2, and 3 hours relative to these peripheral parameters.

The sample size was chosen based on previous studies...”

Reviewer #3 (Remarks to the Author):

1. The authors have carefully revised the manuscript and addressed most of the issues the issues raised by the reviewers. Areas of uncertainty regarding the main claims remain but they are explicitly addressed in the discussion. The statistical analysis has been considerably strengthened. The use of one-tail testing is a weakness but it is clearly acknowledged.

Authors' response:

We would like to mention that we have now decided to report two-sided tests throughout the manuscript because of similar points raised by the other reviewers. Results for cortisol, aldosterone, T cells and B cells do not change by employing two-sided tests, and the effect on prolactin is the only one that now fails to reach statistical significance when clustering the data into one-hour bins. We have now adapted the manuscript accordingly:

Abstract: "Stimulation intensifies the hormonal milieu characterizing SWS (mainly by further reducing cortisol and increasing aldosterone levels) and reduces T and B cell counts, likely reflecting a redistribution of these cells to lymphoid tissues. GH remains unchanged."

Introduction, page 4: "We show that selectively enhancing SOs through auditory stimulation intensifies the immune-supportive hormonal milieu present during SWS (mainly by further reducing cortisol and increasing aldosterone levels) and supports the extravasation of T and B lymphocytes."

Results, page 5: "Aldosterone levels were significantly increased in the Stimulation compared to the Sham condition during the 4th hour after stimulation onset ($p = 0.028$; Fig. 2c, Supplementary Fig. 3b). An increase in aldosterone levels during the 8th hour and in prolactin levels during the 3rd hour after stimulation onset approached significance ($p \leq 0.075$; Fig. 2c,d, Supplementary Fig. 3b,c), with exploratory analyses revealing enhanced prolactin levels after stimulation at two of the 15-min samplings (150 min and 165 min post stimulation onset, respectively, $p \leq 0.050$). GH remained unchanged (Supplementary Fig. 3d)."

Discussion, page 6ff: "We show here that deepening sleep by EEG closed-loop auditory stimulation of SOs robustly decreases levels of the anti-inflammatory hormone cortisol, increases levels of aldosterone and tends to increase also prolactin levels, thus enforcing the immune-supportive hormonal milieu that is unique to SWS³.

... Considering that our method is highly specific in selectively enhancing SOs without affecting other sleep parameters, our findings suggest a causal role of SOs in supporting the overall pro-inflammatory hormonal milieu characterizing SWS.

... The effect of SO stimulation on prolactin levels developed gradually (Supplementary Fig. 3c) and failed to reach significance after clustering data of the frequent blood samplings into one-hour bins.

The pituitary release of prolactin is mainly regulated in a negative, i.e., suppressive manner by dopaminergic activity of the arcuate nucleus of the hypothalamus. Interestingly, this activity exhibits an endogenous rhythm of 0.05-4 Hz²⁹ and, thus, might be sensitive to exogenous stimulation of oscillations in the same frequency range. This idea is supported by our findings. However, the effect of the stimulation on prolactin was not robust, and also in light of conflicting findings as to an association of prolactin release with specific sleep stages²⁰, the role of SOs in the regulation of this hormone needs to be further scrutinized."

Methods, page 16: “Differences between conditions were analyzed with two-sided Student’s t-tests for sleep and EEG data and with two-sided Wilcoxon-signed-rank tests for endocrine and immune parameters as these data were not normally distributed.”

2. The stimulation of SO power and amplitude illustrated in Figure 1 is very modest, under 10% on average. This may be in the range of day to day variability in a given subject. This should also be discussed.

Authors’ response:

The stimulation-induced changes in SO parameters were indeed not very strong, which is not surprising given that we used a very homogenous sample of healthy young men with good sleep. Nevertheless, the size of the effects is comparable to our previous study, in which the same stimulation technique enhanced memory performance (Ngo et al., 2013. *Neuron*). The effect size Cohen’s *d* indicates a medium effect for the SO power ($d = 0.71$) and a small effect for the SO amplitude ($d = 0.42$). We now present the effect sizes and discuss this aspect in the manuscript:

Results, page 5: “... stimulation significantly increased EEG power density in the SO peak frequency (~0.9 Hz, corresponding to the 1.075 s inter-stimulus interval of the applied two tones per stimulation) as well as the amplitude of SOs ($p = 0.006$ and $p = 0.047$, respectively, for electrode position Fz; Fig. 1b,c; Cohen’s $d = 0.71$ and 0.42 , respectively).”

Discussion, page 7: “The lack of effect of the SO stimulation on GH was surprising, given that its release shows a clear association with EEG slow waves and is regulated most strongly by sleep^{6,19,20}. However, the increase in SO activity by the stimulation was only moderate in size (probably due to our subjects being young healthy men with already deep sleep) and, thus, this increase might not have been large enough to affect GH. Nevertheless, the increase in SO activity was comparable in size to that of our previous study¹² and strong enough to affect other parameters. The lack of effect on GH might therefore rather reflect a ceiling effect for GH...”

Methods, page 16: “We calculated the effect size Cohen’s *d* for the impact of the stimulation on SO activity and on the amplitude of SOs, and followed Cohen’s criteria for interpretation of the sizes ($d = 0.2$, small; $d = 0.5$, medium; $d = 0.8$, large)⁵⁷.”

REVIEWERS' COMMENTS:

Reviewer #1 (Remarks to the Author):

The authors have been responsive to my comments and the additional statistical analyses have strengthened the manuscript. The authors have applied these analyses to a 4 hour period which is longer than the two hour interventional period and shorter than the 'recording' time. So again, a somewhat arbitrary post-hoc choice is made. The authors' argument for using 4 hours (this is where most SWS is present) is not convincing at all. If this was so essential why then was the intervention period not 4 hours and why should the effect of a 2 hour intervention, which does not affect SOs in the subsequent hours be limited to the SWS period? In other words, although the additional statistical analyses are informative I remain concerned about the absence of an 'interventional' control.

Reviewer #2 (Remarks to the Author):

In their second revision the authors have addressed the remaining questions. In the reviewer's opinion the manuscript is now more consistent in terms of statistics and the additional statistical analysis helped to clarify the impact of the number of stimuli.

The reviewer was also asked by the editor to comment on the issue of an acoustic stimulation control condition. As this is a newly emerging field, standards for control conditions have not been established. Indeed, establishing such standards would necessitate significant experimentations following a systematic approach to control the effects of acoustic stimuli presented at different phases of the SO on SO power. These experiments are certainly needed but go beyond the scope of this paper. Explicitly stating the need for standards for control conditions might be helpful for the readers and bring up a current limitation of this approach.

Reviewer #3 (Remarks to the Author):

Additional questions arise from the Authors' responses to the previous review.

The Authors' responses to the comments of Reviewer 1 reveal that the stimulation of the "slow oscillations" (SO power; frequency range ≤ 1 Hz) does not necessarily translate into a stimulation of "slow-wave activity" (SWA; frequency range: 0.4 – 4 Hz). Yet, the impact of stimulation of SO power and amplitude on the concomitant activity in the 1-4 Hz range is not presented or discussed. Further, the impact of stimulation of SO power and amplitude on spectral EEG activity in the lower frequency range of SWA during the 2- to 4-hour period after acoustic stimulation (when an impact on aldosterone and perhaps prolactin was detected) is not presented. The responses to the Reviewer and revisions made to the manuscript also include the notion that the narrow SO frequency range may not participate in the control of SWS-related endocrine release which could be dependent on the broader

range of SWA.

All that we know about the endocrine concomitants of slow-wave sleep (SWS) (as described in the introduction of the manuscript) was derived from analyses of simultaneous 30-sec epochs of EEG recording visually scored as SWS and circulating hormonal levels assessed via frequent blood sampling. A vast literature documented that within 5-20 minutes of the initiation of SWS, the release of GH, prolactin and aldosterone is enhanced while cortisol concentrations are suppressed. SWA is consistently elevated during SWS. To my knowledge, there are no studies that attempted to dissociate the very low frequency range of SO from the wider frequency range of SWA in relation to hormonal and immune function. Reference 8 examines the impact of min spent in SWS and SO on immunological memory but does not dissociate SO from SWA. Further, none of the existing studies has documented a delayed impact of SWS/SWA stimulation on hormonal release as is reported in the present manuscript for prolactin and aldosterone.

The data presented in the manuscript show that

1. A modest elevation of SO power and amplitude during stimulation results in a concomitant suppression of cortisol levels during the first hour. No data on the concomitant impact on SWA are presented.
2. A modest elevation of SO power and amplitude during stimulation does not enhance GH release. The possibility that GH release is enhanced by SWA but not by SO power is raised in the revised manuscript but no data on SWA are presented.
3. A modest elevation of SO power and amplitude during stimulation results in an elevation of aldosterone release during the 4th hour after stimulation onset and a trend for increased prolactin release during the 3rd hour after stimulation onset. Profiles of spectral EEG activity during the hours following the end of the stimulation period are not examined.

The sham condition without any acoustic stimulation is in my view an acceptable control condition. Were the participants debriefed after each condition to find out whether they had heard the stimuli and if so, how many ? This should be reported. The mean \pm SEM of normalized SO power and SO amplitude reported in Fig. 2c suggests that there were participants who were "non responders" to acoustic stimulation. This aspect may also need to be elaborated upon.

The Authors embraced the suggestion by Reviewer 1 of a "statistical control" for the number of acoustic stimuli delivered. It seems that the linear regression should have examined "change in the endocrine/immune variables from sham to active condition" versus "change in normalized SO power (or amplitude) from sham to active condition", controlling for the number of stimuli delivered. Instead, the absolute value of the endocrine/immune condition during the active condition was examined in relation to the normalized SO power (or amplitude) of the active condition. It may be that the analysis using "changes" will disclose correlations that were not detected by the analysis of absolute values during the active condition.

In detail, the following changes have been introduced regarding the referees' comments (highlighted in yellow):

Reviewer #1 (Remarks to the Author):

The authors have been responsive to my comments and the additional statistical analyses have strengthened the manuscript. The authors have applied these analyses to a 4 hour period which is longer than the two hour interventional period and shorter than the 'recording' time. So again, a somewhat arbitrary post-hoc choice is made. The authors' argument for using 4 hours (this is where most SWS is present) is not convincing at all. If this was so essential why then was the intervention period not 4 hours and why should the effect of a 2 hour intervention, which does not affect SOs in the subsequent hours be limited to the SWS period? In other words, although the additional statistical analyses are informative I remain concerned about the absence of an 'interventional' control.

Authors' response:

As indicated in the Methods section, the correlations were not significant when they were restricted to the stimulation period. This is very likely due to the low variance in SO activity during the stimulation, producing close to ceiling SO activity. We therefore decided to extend the analyzed interval in order to increase the variability between subjects. We understand that selecting the 4 hour interval might seem to some extent arbitrary. However, selecting the time window with predominant SWS makes absolute sense (rather than e.g. selecting the entire sleep period) because these analyses, in the first place, aimed at scrutinizing the link between SOs and the hormonal and immunological measures of interest, and SWS is the sleep stage during which most (induced as well as spontaneous) SOs occurred.

Reviewer #2 (Remarks to the Author):

In their second revision the authors have addressed the remaining questions. In the reviewer's opinion the manuscript is now more consistent in terms of statistics and the additional statistical analysis helped to clarify the impact of the number of stimuli.

The reviewer was also asked by the editor to comment on the issue of an acoustic stimulation control condition. As this is a newly emerging field, standards for control conditions have not been established. Indeed, establishing such standards would necessitate significant experimentations following a systematic approach to control the effects of acoustic stimuli presented at different phases of the SO on SO power. These experiments are certainly needed but go beyond the scope of this paper. Explicitly stating the need for standards for control conditions might be helpful for the readers and bring up a current limitation of this approach.

Authors' response:

We agree with the reviewer that it is important to include a more explicit statement on the need for developing more control conditions. We adapted the discussion accordingly:

Discussion, page 11: "... To test whether the effects are specific for the closed-loop stimulation one might think of a control group stimulated in a random or open-loop fashion, i.e., by applying the same auditory stimuli during NonREM sleep independently of the endogenous brain rhythm.

However this approach is not necessarily effective: A previous study using this open-loop stimulation to test its effects on memory did not find a functional improvement in memory, but still an increased SO activity (although less robustly) after open-loop stimulation⁴⁹. In addition, this random application of stimuli during NonREM sleep changed other sleep parameters and was noticed during the night by half of the subjects, thus rendering such stimulation an inappropriate control⁴⁹. In the current study, we therefore employed sham stimulation as the most “blank” control condition. Nevertheless, it remains a challenge to future studies to establish control conditions that enable to experimentally dissociate the effects of our closed-loop stimulation from those of the auditory stimulus presentation per se. This means, further efforts should be undertaken to establish a control condition of auditory stimulation that would not induce SOs or even robustly and selectively suppress them.”

Reviewer #3 (Remarks to the Author):

Additional questions arise from the Authors’ responses to the previous review.

The Authors’ responses to the comments of Reviewer 1 reveal that the stimulation of the “slow oscillations” (SO power; frequency range ≤ 1 Hz) does not necessarily translate into a stimulation of “slow-wave activity” (SWA; frequency range: 0.4 – 4 Hz). Yet, the impact of stimulation of SO power and amplitude on the concomitant activity in the 1-4 Hz range is not presented or discussed. Further, the impact of stimulation of SO power and amplitude on spectral EEG activity in the lower frequency range of SWA during the 2- to 4-hour period after acoustic stimulation (when an impact on aldosterone and perhaps prolactin was detected) is not presented. The responses to the Reviewer and revisions made to the manuscript also include the notion that the narrow SO frequency range may not participate in the control of SWS-related endocrine release which could be dependent on the broader range of SWA.

All that we know about the endocrine concomitants of slow-wave sleep (SWS) (as described in the introduction of the manuscript) was derived from analyses of simultaneous 30-sec epochs of EEG recording visually scored as SWS and circulating hormonal levels assessed via frequent blood sampling. A vast literature documented that within 5-20 minutes of the initiation of SWS, the release of GH, prolactin and aldosterone is enhanced while cortisol concentrations are suppressed. SWA is consistently elevated during SWS. To my knowledge, there are no studies that attempted to dissociate the very low frequency range of SO from the wider frequency range of SWA in relation to hormonal and immune function. Reference 8 examines the impact of min spent in SWS and SO on immunological memory but does not dissociate SO from SWA. Further, none of the existing studies has documented a delayed impact of SWS/SWA stimulation on hormonal release as is reported in the present manuscript for prolactin and aldosterone.

The data presented in the manuscript show that

- 1. A modest elevation of SO power and amplitude during stimulation results in a concomitant suppression of cortisol levels during the first hour. No data on the concomitant impact on SWA are presented.*
- 2. A modest elevation of SO power and amplitude during stimulation does not enhance GH release. The possibility that GH release is enhanced by SWA but not by SO power is raised in the revised manuscript but no data on SWA are presented.*
- 3. A modest elevation of SO power and amplitude during stimulation results in an elevation of*

aldosterone release during the 4th hour after stimulation onset and a trend for increased prolactin release during the 3rd hour after stimulation onset. Profiles of spectral EEG activity during the hours following the end of the stimulation period are not examined.

Authors' response:

We thank the reviewer for raising the important issue of discriminating between SO and SWA. The studies mentioned in the introduction of our manuscript mostly investigated the entire SWA range (0.5-4 Hz) and as such did not differentiate between the SO frequency range and the 1-4 Hz delta wave range. (Also, in most of these studies the authors referred to the entire SWA frequency range as "delta", which can create further confusion). These studies therefore do not provide information on which frequency within the SWA range is most important for the associations between SWA and the hormones of interest. In contrast, the mentioned reference 8 (Lange et al., 2011) indeed focused on the SO frequency range when analyzing the association with the immune response (see Methods section of that paper). This is the main reason why we decided to focus the stimulation on the SO frequency in the current study (apart from the methodology being already well-established in our lab). Based on our data alone, we are hesitant to go further into a discussion of potential functional differences between the SO and higher SWA frequencies. However, please note that we do show in the results (see Supplementary Figure 1) that the delta wave range was not changed by the stimulation and mention this also in the discussion.

The sham condition without any acoustic stimulation is in my view an acceptable control condition. Were the participants debriefed after each condition to find out whether they had heard the stimuli and if so, how many? This should be reported. The mean \pm SEM of normalized SO power and SO amplitude reported in Fig. 2c suggests that there were participants who were "non responders" to acoustic stimulation. This aspect may also need to be elaborated upon.

Authors' response:

The participants were asked after each condition if they had heard any tones during the night, and the number of subjects that reported having noticed any tones (i.e. two out of 14 subjects) is mentioned in the Methods section (on page 13). Subjects were not debriefed after the first condition in order to avoid unblinding them for the second condition. Responsiveness to the tones was confirmed by inspecting the auditory evoked potential response from each subject. We now added this information to the manuscript as follows:

Methods, page 15: "Responsiveness of subjects to the stimulation was confirmed by measuring the individual's averaged evoked potential response to the tones."

The Authors embraced the suggestion by Reviewer 1 of a "statistical control" for the number of acoustic stimuli delivered. It seems that the linear regression should have examined "change in the endocrine/immune variables from sham to active condition" versus "change in normalized SO power (or amplitude) from sham to active condition", controlling for the number of stimuli delivered. Instead, the absolute value of the endocrine/immune condition during the active condition was examined in relation to the normalized SO power (or amplitude) of the active condition. It may be that the analysis using "changes" will disclose correlations that were not detected by the analysis of absolute values during the active condition.

Authors' response:

Calculating the changes between conditions did not reveal any significant correlations. This is probably again due to the fact that stimulation enhanced SO activity close to a physiological ceiling effect, resulting in low effective variance. Moreover, such change measures can be considered more “noisy” because error variance from the sham condition is added. We think that presenting correlations for the absolute values is most informative here because these values reflect the physiological associations between SOs (whether induced or spontaneous) and the hormones. Most importantly, the regression analyses we employed allowed us to directly control for the number of applied stimuli, and this was essential for statistically excluding that the stimuli *per se* induced the observed changes (as requested by Reviewer 1).